# GUI-Shepherd: Reliable Process Reward and Verification for Long-Sequence GUI Tasks

## Abstract

Autonomous agents for long-sequence Graphical User Interface tasks are hindered by sparse rewards and the intractable credit assignment problem. To address these challenges, we introduce GUI-Shepherd, a Process Reward Model that provides dense, step-by-step feedback to guide agents. GUI-Shepherd is trained on a diverse large-scale data set of 52k interactions that features human-annotated scores and GPT-4o generated rationales, enabling it to serve both as a reward provider for RL training and as a verifier for inference. As far as we know, we are the first to conduct a systematic study of process supervision in GUI agents, across diverse settings from online long-horizon tasks to offline single-step prediction. On the online AndroidWorld benchmark, GUI-Shepherd improves success rate by 7.7 points via multi-turn online PPO, significantly outperforming Outcome Reward Model based competitors. When used as an inference verifier, it brings 5.1 points improvements. The benefits generalize to the offline AndroidControl benchmark, with gains of 2.2 points as a reward provider and 4.3 points as a verifier. Collectively, our results establish that high-fidelity process supervision is critical for building more capable GUI agents and present a generalizable solution.

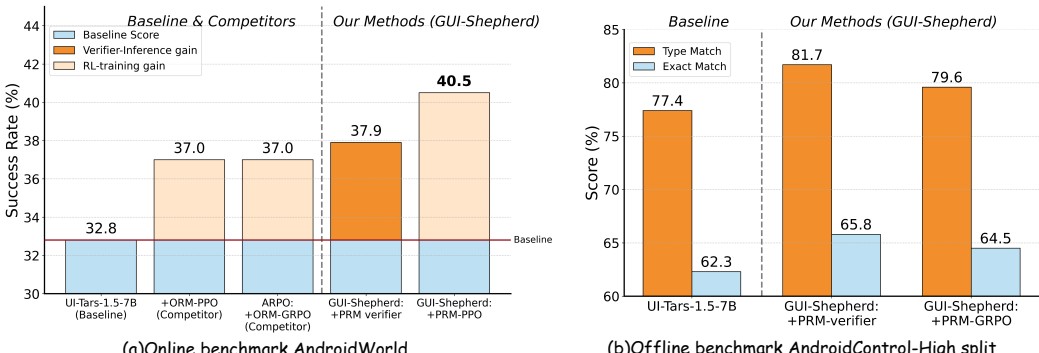

Figure 1: **GUI-Shepherd significantly enhances agent performance in both online long-sequence and offline single-step settings.** (a) On the online AndroidWorld, our PRM-guided PPO agent achieves a $40.5\%$ success rate ($+7.7$ over the $32.8\%$ baseline), outperforming ORM-based competitors ($37.0\%$). As a standalone verifier, it boosts performance to $37.9\%$ ($+5.1$). (b) This advantage extends to the offline AndroidControl benchmark, where as a verifier, GUI-Shepherd improves Type Match to $81.7\%$ ($+4.3$), and as a reward provider, it improves Type Match to $79.6\%$ ($+2.2$).

## 1 Introduction

Developing agents(Zhang et al., 2025a; Zhang & Zhang, 2023) to execute human instructions via Graphical User Interface (GUI) is a critical AI frontier. Although Large Vision-Language Models(Bai et al., 2025; Chen et al., 2024; Liu et al., 2023) provide the foundational perceptual and reasoning capabilities for such agents, their practical utility is severely hampered by poor performance on

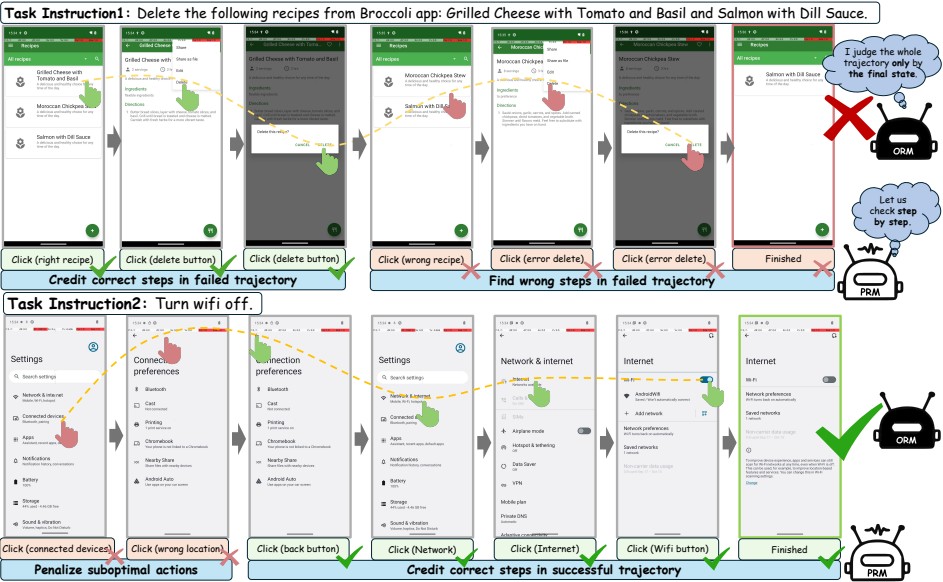

Figure 2: Process-based vs. Outcome-based reward models. An ORM judges a trajectory by its final state, failing to **identify critical errors, assign credit to correct steps, or penalize suboptimal actions.** While a PRM evaluates each action individually, it provides a more accurate signal.

long-horizon tasks(Xu et al., 2024a). This limitation is stark: despite excelling at single-step actions, these agents frequently fail complex long-sequence workflows.

The iterative and stochastic nature of long-horizon GUI tasks necessitates robust planning and reasoning beyond the rote memorization of expert trajectories. This presents a challenge for Supervised Fine-Tuning, which is constrained by both the practical difficulty of scaling vast data requirements and the conceptual limitation that imitation may be an inefficient path toward robust reasoning. While Reinforcement Learning(Rafailov et al., 2023; Shao et al., 2024; Guo et al., 2025) offers a theoretically aligned framework for long-horizon planning, its application to GUI tasks(Lu et al., 2025a) has been hindered by the sparse signals from conventional Outcome Reward Model(ORM). As illustrated in Figure 2, an ORM provides a single terminal signal that fails to **identify critical errors, reward correct steps on failed trajectories or penalize suboptimal actions**.

The limitations of outcome-based signals in the GUI domain motivate a shift towards denser, more informative feedback. We therefore adopt a fundamentally different approach: Process-based Reward Models (PRMs)(Luo et al., 2024), which provide step-by-step feedback and have been proven effective in complex math reasoning tasks(Lightman et al., 2023). Our paper presents a systematic investigation into the training of a robust PRM for GUI agents, and comprehensively validates its efficacy across a wide range of agent workflows: online/offline RL and inference-time verification.

To construct a reliable PRM, we develop a dual-pipeline methodology to curate a diverse 52k-sample dataset, balancing two critical axes: **temporal and UI diversity**. To achieve temporal diversity, we generate full trajectories in an interactive environment to capture the varied states across all stages of long-horizon tasks. Concurrently, to enrich the diversity of UI, we sample single-step states from a highly diverse dataset(Li et al., 2024) to ensure a vast range of applications and UI layouts. Our annotation process revealed a crucial insight: even sota VLM like GPT-4o(Hurst et al., 2024) still exhibits a significant performance gap compared to human experts, underscoring the necessity of human annotation for the core correctness labels. We therefore adopt a hybrid strategy to balance quality and cost: human annotators provide the high-reliability binary correctness scores, while GPT-4o generates the explanatory chain-of-thought reasoning.

We validate GUI-Shepherd's impact in the most demanding setting: **as a dense reward provider for online RL in a dynamic environment, a challenging frontier that remains largely unexplored**. When integrated with Proximal Policy Optimization (PPO) algorithm(Schulman et al., 2017) on the AndroidWorld benchmark, GUI-Shepherd guides the agent to 7.7 points improvement in success rate.

As shown in Figure 1, this result significantly outperforms strong ORM-based competitors by 3.5 points, providing strong empirical evidence for the superiority of process-based rewards in complex interactive tasks. Furthermore, when deployed as an inference-time verifier to re-rank candidate actions, GUI-Shepherd boosts the base agent by 5.1 points, demonstrating its dual utility.

To establish that these benefits stem from a general principle rather than a specialized solution for long-horizon tasks, we test GUI-Shepherd on single-step action prediction tasks on an offline dataset, *i.e.*, AndroidControl(Li et al., 2024), which consists of pre-collected trajectories. GUI-Shepherd again proves its versatility, improving Type Match of High Split as both an inference time verifier (+4.3) and an offline RL reward provider (+2.2). This success across different domains substantiates our central thesis: step-by-step process supervision is a powerful and broadly applicable paradigm for advancing GUI automation.

In summary, our contributions are:

1. We are the first to successfully apply PRM to online RL in long-horizon GUI tasks, tackling sparse reward problem and achieving a significant 7.7 point improvement in success rate.

2. We further present systematic validation of PRM in the GUI domain, from online long-sequence task completion to single-step action prediction, demonstrating its effectiveness as both a reliable reward provider for online/offline RL, and a robust verifier for inference.

3. We introduce a novel and scalable dual-pipeline for creating high-quality process supervision datasets, addressing the critical data prerequisite for this line of research by ensuring both temporal and UI diversity.

## 2 RELATED WORK

### 2.1 GUI AGENTS

The field of GUI agents has rapidly evolved from non-visual models that processed structured inputs like XML(Guo et al., 2024) to modern agents based on VLMs that perceive raw screen pixels(Cheng et al., 2024). Methodologically, the paradigm has shifted from Supervised Fine-Tuning, which focuses on fundamental grounding skills(Hong et al., 2024), towards reinforcement learning(Bai et al., 2024) to improve decision-making. A significant body of work has demonstrated success in training agents for single-step actions, typically using offline RL on offline datasets with rule-based rewards(Luo et al., 2025; Lu et al., 2025b). However, this success has been largely confined to single-step or short-horizon settings. The majority of current methods exhibit a sharp performance degradation when faced with complex, long-sequence tasks. This represents a critical limitation, as such tasks are the most practical and challenging use case for GUI agents. Our work directly confronts this challenge by focusing on enhancing agent reliability in these long-sequence scenarios.

### 2.2 OFFLINE VS. ONLINE EVALUATION FOR GUI AGENTS

Evaluation benchmarks for GUI agents can be categorized by single-step action prediction and end-to-end task completion. A larger body of work focuses on single-step action evaluation, which is typically conducted offline using offline datasets of expert trajectories(Rawles et al., 2023; Zhang et al., 2024; Lu et al., 2024). This paradigm, where an agent's prediction is compared against a ground-truth action, is well-suited for the supervised and offline RL methods. However, evaluating an agent's ability to complete long-sequence tasks is substantially more complex, necessitating dynamic, interactive environments that respond to agent actions in real-time. This has led to the development of sophisticated online benchmarks like OSWorld(Xie et al., 2024) for desktop and AndroidWorld(Rawles et al., 2024) for mobile platforms. It is in these challenging, long-horizon environments that current agents falter. Our work is therefore focused on improving task success rates on AndroidWorld, which comprehensively evaluates the critical, end-to-end capabilities of agents.

### 2.3 PROCESS REWARD MODELING

Process Reward Modeling(Luo et al., 2024; Khalifa et al., 2025) has become a key technique for overcoming the sparse reward and credit assignment challenges in complex, multi-step reasoning

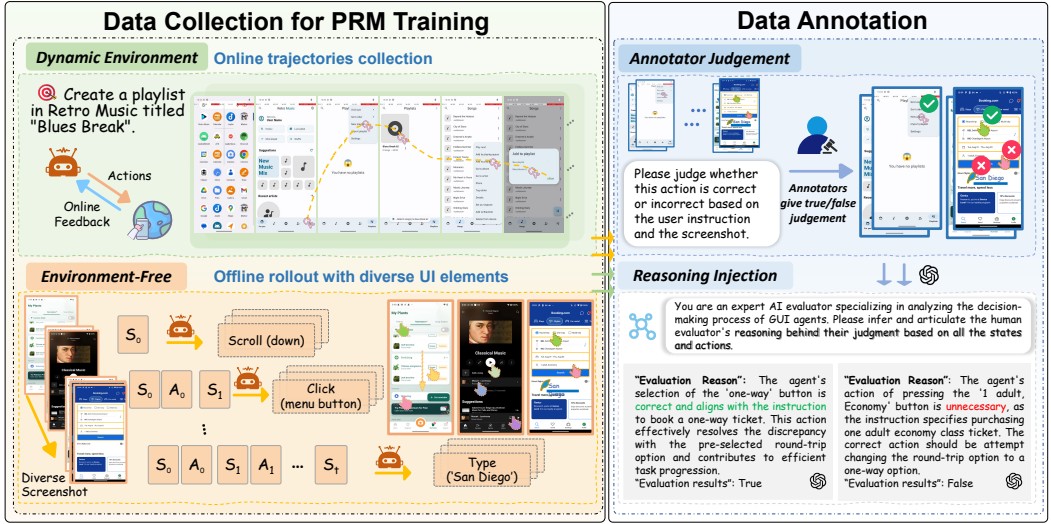

Figure 3: Our data curation pipeline for the PRM. (Left) A dual data collection pipeline combines rollout online trajectories and offline samples. (Right) A hybrid annotation process where humans provide binary correctness labels, which are then augmented with GPT-generated chain-of-thought.

tasks. Its efficacy is well-established in domains like mathematics, where PRMs serve a dual role: providing dense rewards for reinforcement learning and verifying steps at inference time(Wang et al., 2023; Lightman et al., 2023; Ma et al., 2023; Zhu et al., 2025). Inspired by this success, we investigate the application of PRMs to GUI agents. While existing GUI agent research has touched upon related concepts, such as learning critic(Wanyan et al., 2025; Hu et al., 2025) or value models(Zheng et al., 2025) to provide richer training signals or help refine action prediction, the systematic development and application of a dedicated PRM remains an open area. This paper presents the first comprehensive framework for process supervision in GUI agents, spanning from data collection to downstream deployment for online RL training, inference-time verification, and offline RL training.

## 3 TRAINING THE SHEPHERD: A RELIABLE PRM

A long-sequence GUI task with a high-level instruction $I$ requires an agent to execute a trajectory of state-action pairs, $\tau = (s_0, a_0, \ldots, s_T, a_T)$, where $T$ is the number of steps, $s_t$ is the screen screenshot at time step $t$, and $a_t$ is a GUI operation (*e.g.*, click, type). Standard approaches that rely on an ORM to assign a single, terminal reward to the entire trajectory are limited by reward sparsity, which provides inadequate supervision for intermediate steps. To address this, we introduce a Process Reward Model that provides dense, step-wise supervision. Specifically, our PRM is formulated as a function that assesses the correctness of an action $a_t$ at a given state $s_t$ conditioned on the instruction $I$. It outputs a binary classification label, $r_t \in \{\text{positive}, \text{negative}\}$, indicating the quality of the step:

$$r_t = \mathcal{R}_{\text{PRM}}(I, s_t, a_t) \tag{1}$$

Building a reliable PRM capable of providing such supervision is a primary challenge. We establish a systematic methodology of three key pillars: data preparation, annotation paradigms, and training.

**Training Data Preparation.** The training data for our PRM consists of quadruplets, $(I, s, a, r)$. We employ a dual-pipeline collection strategy to balance **temporal and UI diversity** as shown in Figure 3. The first pipeline sources temporal diversity by executing full trajectory rollouts with agents in Android emulators. While this method is time-consuming and limited in application diversity, it is crucial for capturing the varied, in-distribution states that occur across long-horizon tasks. To complement this with UI diversity, we sample single-step states from the large-scale AndroidControl(Li et al., 2024) training set and then performs one-step rollouts to gather interaction data across wider applications and UI layouts. The fusion of these two data sources creates a varied dataset representative of real agent behavior, ultimately yielding 52k training set.

**Reward Annotation.** Each data point $(I, s, a)$ in our dataset is annotated with a binary score $r$(positive or negative) indicating the correctness of the action $a$. Empirically, binary scores provided by human annotators are more reliable than those generated by sota proprietary models such as GPT-4o. Therefore, to ensure the highest quality for our primary supervisory signal, all binary scores in our dataset are human-annotated. To manage annotation costs, we then use the proprietary model GPT-4o to generate the corresponding chain-of-thought rationales. We also perform manual spot checks on annotated data to ensure its integrity further. These steps, which combine human expert judgment for the core scores with rigorous quality control, ensure the reliability of our final dataset.

**Training Strategy.** We initialize our PRM from policy model UI-TARS-1.5-7B(Qin et al., 2025) and train the PRM via SFT. To represent the binary correctness labels, we map positive and negative to two specific tokens in the model's vocabulary, which allows us to train the PRM for classification without requiring any architectural modifications. The model, parameterized by $\theta$, is trained to predict the ground-truth label token $y$ with a standard cross-entropy loss over these two tokens:

$$\mathcal{L}_{\text{SFT}}(\theta) = -\log P_\theta(y|I, s, a) \tag{2}$$

To investigate the effectiveness of CoT reasoning, we propose a variant that first generates a CoT rationale before outputting the final score. Our analysis shows that generating a reasoning chain helps the model arrive at a more accurate final judgment, and yields a more accurate and reliable PRM.

## 4 Guiding the Agent: Training and Verification

We utilize our trained Process Reward Model to significantly improve agent performance on online long-sequence tasks by providing both per-step rewards for RL training and per-step verification for inference. We further show that its advantages extend to offline single-action prediction.

### 4.1 PRM for Online Reinforcement Learning

The virtual environments in AndroidWorld require direct access to physical hardware, making them incompatible with containerized cluster environments. To overcome this, we refactored the interaction layer with the Android emulator and reimplemented the evaluation logic, enabling agents to interact with remote virtual emulators via a standardized IP and port interface. To validate our implementation, we benchmark the UI-TARS-1.5-7B baseline in our environment, achieving a $32.8\%$ success rate, consistent with results from community reproductions and confirming the fidelity of our setup.

Building on our robust interactive environment, we use UI-TARS-1.5-7B as the policy $\pi_\theta$ to implement an online, multi-turn PPO. Crucially, we leverage our pre-trained PRM to initialize the weights of the value model $V_\phi$. As detailed in Algorithm 1, our online training begins with parallelized trajectory collection from multiple remote Android emulators. For each step within these trajectories, we compute a dense reward by querying a PRM service deployed on a separate vLLM(Kwon et al., 2023) node and incorporating a format-based reward. The total reward is thus formulated as:

$$r_t = w_p \cdot \mathcal{R}_{\text{PRM}}(I, s_t, a_t) + w_f \cdot \mathcal{R}_{\text{format}}(a_t) \tag{3}$$

This dense, per-step reward is pivotal for calculating the advantage via GAE(Schulman et al., 2015):

$$A_t^{\text{GAE}} = \sum_{k=0}^{T-t-1} (\gamma\lambda)^k (r_{t+k} + \gamma V_\phi(s_{t+k+1}) - V_\phi(s_{t+k})) \tag{4}$$

By infusing the reliable PRM score ($\mathcal{R}_{\text{PRM}}(I, s_t, a_t)$) into each $r_t$, GAE produces a fine-grained advantage estimate. This allows it to accurately **assign credit or blame to individual actions within a long trajectory**, effectively guiding the policy's gradient updates. The policy $\pi_\theta$ is subsequently updated by maximizing the PPO-Clip objective function:

$$\mathcal{L}^{\text{CLIP}}(\theta) = \hat{\mathbb{E}}_t \left[ \min\left( \frac{\pi_\theta(a_t|s_t)}{\pi_{\theta_{\text{old}}}(a_t|s_t)} A_t, \text{clip}\left( \frac{\pi_\theta(a_t|s_t)}{\pi_{\theta_{\text{old}}}(a_t|s_t)}, 1 - \epsilon, 1 + \epsilon \right) A_t \right) \right] \tag{5}$$

As shown in Table 1, this PRM-guided PPO approach achieves 7.7 absolute points improvement in success rate on AndroidWorld. To rigorously evaluate this gain, we also implemented two baselines: an ORM-based multi-turn PPO and a reproduction of the ORM-based trajectory-level GRPO algorithm ARPO(Lu et al., 2025a). Our results confirm that the dense, reliable reward from the PRM significantly enhances agent performance compared to reward-sparse ORM.

---

**Algorithm 1** PRM-based Online PPO

1: **Initialize:** Policy $\pi_\theta$, value model $V_\phi$, PRM $\mathcal{R}_{\text{PRM}}$, emulators $\{E_i\}_{i=1}^N$, optimizer Adamw.
2: **for** iteration = 1, 2, ... **do**
3: $\quad \mathcal{D} \leftarrow \bigcup_{i=1}^N \text{Rollout}(\pi_\theta, E_i)$
4: $\quad$ **for** each step $(I, s_t, a_t) \in \mathcal{D}$ **do**
5: $\quad\quad r_t \leftarrow w_p \cdot \mathcal{R}_{\text{PRM}}(I, s_t, a_t) + w_f \cdot \mathcal{R}_{\text{format}}(a_t)$
6: $\quad$ **end for**
7: $\quad \{\hat{A}_t\} \leftarrow \text{GAE}(\{r_t\}, \{V_\phi(s_t)\})$
8: $\quad \theta \leftarrow \text{Adamw}(\theta, \nabla_\theta \mathcal{L}^{\text{CLIP}}(\theta; \hat{A}_t))$
9: $\quad \phi \leftarrow \text{Adamw}(\phi, \nabla_\phi \mathcal{L}^{\text{VF}}(\phi))$
10: **end for**

---

**Algorithm 2** PRM-based Verification

1: **Inputs:** Policy $\pi_\theta$, PRM $\mathcal{R}_{\text{PRM}}$, state $s_t$, instruction $I$, num candidates $n$.
2: $A_c = \{a_1, \ldots, a_n\} \sim \pi_\theta(\cdot|s_t, I)$
3: $\text{L} \leftarrow [\,]$
4: **for** each action $a_i \in A_c$ **do**
5: $\quad l_i \leftarrow \text{Logits}(\mathcal{R}_{\text{PRM}}(I, s_t, a_i))_{\text{positive}}$
6: $\quad$ Append $l_i$ to L
7: **end for**
8: $i^* \leftarrow \arg\max_i \text{L}$
9: **return** $a_{i^*}$

---

## 4.2 PRM FOR INFERENCE-TIME VERIFICATION

In addition to playing a reward provider role in training, we also leverage GUI-Shepherd as a verifier to enhance action selection at inference time, as shown in Algorithm 2. Instead of greedily selecting a single action from the agent policy $\pi_\theta$, we employ a candidate re-ranking strategy. At each decision step $t$, we first sample a set of $n$ candidate actions $\mathcal{A}_t = \{a_{t,1}, \ldots, a_{t,n}\}$ from the policy's distribution. Each candidate action $a \in \mathcal{A}_t$ is then scored by GUI-Shepherd. Specifically, we use the PRM's output logit corresponding to "positive" as the verification score, denoted as $\text{L}_{\text{pos}}(I, s_t, a)$. The agent then executes the action $a_t^*$ with the highest score:

$$a_t^* = \arg\max_{a \in \mathcal{A}_t} \text{L}_{\text{pos}}(I, s_t, a) \tag{6}$$

This verification process is repeated at each step to construct the full task trajectory. This approach, as a test-time verifier, improves the base policy without any modifications to its parameters.

## 4.3 PRM FOR OFFLINE REINFORCEMENT LEARNING

Furthermore, we wonder if the advantages that our PRM demonstrated in long-sequence tasks can generalize to offline, single-step action prediction. For the offline RL algorithm, we select GRPO(Shao et al., 2024) for its implementation simplicity and strong performance. Our method involves generating a group of candidate actions for a given state, using our PRM to compute a reward for each action, and then calculating the advantage across this group to update the policy. Separately, we also utilize GUI-Shepherd as an inference-time verifier for single-step action prediction. The results shown in Table 2 are compelling: using PRM-generated rewards for offline GRPO training yields 2.2 points improvement, while deploying it as a verifier at inference time boosts performance by 4.3 points. This demonstrates our PRM's reliable process supervision extends from long-sequence GUI tasks to general single-action prediction.

## 5 EXPERIMENTS

**Implementation Details.** We use UI-TARS-1.5-7B(Qin et al., 2025) as our baseline actor and initialization for our PRM. The PRM training dataset comprises 52k samples, consisting of 26k interactions from the AndroidWorld online environment and 26k from the training set of Android-Control. This dataset maintains a balanced $1:1$ ratio of positive to negative examples. The PRM is SFT trained for 2 epochs to predict final binary scores with reasoning processes. For online RL experiments, we implement multi-turn PPO(Schulman et al., 2017) agents that interact with Android emulators. For offline RL experiments, we implement a standard GRPO(Shao et al., 2024) algorithm. Due to page limits, more training details are provided in the Appendix C and Appendix D.

**Benchmarks and Baselines.** To evaluate agent performance on long-sequence tasks, we use **AndroidWorld**(Rawles et al., 2024). This benchmark evaluates an agent's success rate (SR)(Vattikonda et al., 2025) on 116 multi-step tasks within an online environment, with success determined by programmatic checks. On this benchmark, we compare our PRM-guided agent against two strong

| Method | Online RL | Verifier | AndroidWorld SR (%) ↑ |
|---|:---:|:---:|:---:|
| *State-of-the-art Methods* | | | |
| InfiGUIAgent-2B(Liu et al., 2025) | ✗ | ✗ | 9 |
| Qwen2.5-VL-7B(Bai et al., 2025) | ✗ | ✗ | 22.0 |
| EcoAgent(Yi et al., 2025) | ✗ | ✗ | 27.6 |
| Qwen2.5-VL-72B(Bai et al., 2025) | ✗ | ✗ | 35.0 |
| MobileGUI-7B(Shi et al., 2025) | GRPO | ✗ | 30.0 |
| GUI-Critic-R1(Wanyan et al., 2025) | ✗ | ✓ | 27.6 |
| *On the UI-TARS-1.5-7B Baseline* | | | |
| UI-TARS-1.5-7B(Qin et al., 2025) | ✗ | ✗ | $32.8(\pm0.89)$ |
| +ARPO(Lu et al., 2025a) | GRPO | ✗ | $37.0(\pm0.55)$ |
| +ORM-based PPO | PPO | ✗ | $37.0(\pm0.71)$ |
| **+ PRM-based PPO (Ours)** | PPO | ✗ | $\mathbf{40.5}(\pm0.63)$ |
| **+ PRM verifier (Ours)** | ✗ | ✓ | $\mathbf{37.9}(\pm0.55)$ |

Table 1: Comparison on the AndroidWorld benchmark. GUI-Shepherd enhances the strong baseline. As a reward provider for PPO, it achieves a $40.5\%$ success rate, outperforming the ORM-based approach ($37.0\%$). As an inference-time verifier, it boosts the baseline performance by 5.1 (37.9%).

baselines: (1) an ORM-based PPO agent that relies on a sparse, terminal reward, and (2) our implementation of ARPO(Lu et al., 2025a), an ORM-based GRPO algorithm originally developed for the OSWorld(Xie et al., 2024) desktop environment. To assess the PRM's potential in offline settings, We use the **AndroidControl** benchmark(Li et al., 2024), specifically targeting Type Match (matching the action category, e.g., click, type) and Exact Match (matching the complete action, including category, bounding box, and text input) on the high-level split. This split is particularly challenging as it omits step instructions, compelling the agent to rely on its own reasoning to predict the correct action.

## 5.1 MAIN RESULTS

**Effectiveness in Online RL.** We first evaluate GUI-Shepherd's ability to improve agent performance on long-sequence GUI tasks by providing a dense and reliable reward signal. Our primary experiment integrates the PRM as a step-level reward provider with PPO on the challenging Android-World benchmark. To rigorously validate the superiority of our process-based rewards, we establish strong baselines by utilizing our own ORM-based PPO implementation and reproducing ARPO, an ORM-based GRPO algorithm originally designed for desktop environments. As shown in Table 1, our PRM-guided PPO achieves a significant 7.7 absolute points improvement in success rate over the baseline, and substantially outperforms the ORM-based counterparts. These results strongly indicate that GUI-Shepherd serves as a superior reward provider; its ability to generate a dense and reliable reward signal is critical for effective credit assignment, successfully guiding the agent to learn complex, long-horizon policies where conventional outcome-based rewards fall short.

**Effectiveness of Inference-Time Verification.** Beyond its utility in RL training, we demonstrate that GUI-Shepherd also functions as a highly effective inference-time verifier. During inference, we task the agent with generating a set of candidate actions with $n = 3$ at each step. The PRM then scores each candidate and select the action corresponding to the highest logit for the positive token for execution. The results, detailed in Table 1, are compelling: by applying this verification mechanism to the base agent without any additional training, we achieve a substantial 5.1 point improvement in success rate. This finding is significant, as it indicates the PRM serves as a crucial corrective mechanism, steering the agent away from plausible but incorrect actions and thereby stabilizing its reasoning during long-horizon tasks. Furthermore, it validates inference-time verification as a general and computationally efficient method for enhancing the performance and reliability of GUI agents.

**Effectiveness in Offline Task.** To substantiate that the benefits of our PRM stem from a fundamental principle of process supervision, rather than a specialized solution for long-horizon tasks, we evaluate its efficacy in an offline, single-step action prediction setting on the AndroidControl benchmark. We test GUI-Shepherd in two distinct roles as shown in Table 2. First, when used as a reward

| Model | AC High | | AC Low | |
|---|---|---|---|---|
| | TM (%)↑ | EM (%)↑ | TM (%)↑ | EM (%)↑ |
| *State-of-the-art Methods* | | | | |
| Qwen2.5-VL-7B(Bai et al., 2025) | 69.7 | 57.4 | 92.1 | 82.1 |
| OS-Genesis-7B(Sun et al., 2024) | 65.9 | 44.4 | 90.7 | 74.2 |
| OS-Atlas-7B(Wu et al., 2024) | 70.4 | 56.5 | 73.0 | 67.3 |
| Aguvis-7B(Xu et al., 2024b) | 65.6 | 54.2 | 93.9 | 89.4 |
| OdysseyAgent(Liu et al., 2024) | 58.8 | 32.7 | 65.1 | 39.2 |
| GUI-R1-7B(Luo et al., 2025) | 71.6 | 51.7 | 85.2 | 66.5 |
| AgentCPM-GUI(Zhang et al., 2025b) | 77.7 | **69.2** | 94.4 | **90.2** |
| *On the UI-TARS-1.5-7B Baseline* | | | | |
| UI-TARS-1.5-7B(Qin et al., 2025) | 77.4($\pm$0.22) | 62.3($\pm$0.19) | 95.3($\pm$0.20) | 87.0($\pm$0.21) |
| **+PRM-GRPO (Ours)** | 79.6($\pm$0.18) | 64.5($\pm$0.18) | **95.9**($\pm$0.18) | 87.6($\pm$0.19) |
| **+PRM-verifier (Ours)** | **81.7**($\pm$0.20) | 65.8($\pm$0.19) | 95.4($\pm$0.18) | 87.0($\pm$0.18) |

Table 2: Results for single-step action prediction on the offline AndroidControl benchmark: GUI-Shepherd shows consistent advantages both as a reward provider for offline GRPO, improving High split Type Match to 79.6%, and as an inference-time verifier, boosting the baseline to 81.7%.

| Annotator | Annotation. Acc. (%) ↑ | PRM Acc. (%) ↑ | AW SR (%) ↑ |
|---|---|---|---|
| *Baseline Agent* | | | |
| UI-TARS-1.5-7B | - | - | 32.8 |
| *Agent with PRM Verifier trained by:* | | | |
| GPT-4o (Base Prompt) | 86 | 86.3 | 33.6 |
| GPT-4o (Improved Prompt) | 92 | 89.7 | 34.4 |
| Human Expert | **98** | **94.9** | **36.2** |

Table 3: Results for annotation quality impact on AndroidWorld benchmark. Higher-quality annotation data yields a more accurate PRM, which significantly improves the agent's success rate on long-sequence GUI tasks by providing more reliable verification.

provider for an offline GRPO algorithm, it yields a 2.2 points performance gain, demonstrating its utility for offline policy refinement. Second, when deployed as an inference-time verifier to rank $n = 5$ candidate actions, the PRM produces a more pronounced improvement on the benchmark's High split, boosting Type Match by 4.3 points and Exact Match by 3.5 points over the baseline. The success of GUI-Shepherd in these varied settings confirms that reliable, process-based reward is a versatile and broadly applicable paradigm for enhancing GUI agents, whether for complex, long-sequence tasks or discrete, single-step actions.

## 5.2 Ablation Study and Analysis

**Impact of Annotation Quality.** We conduct an ablation study to analyze the impact of annotation quality on the PRM's ability to provide reliable supervision. We compare three distinct annotation sources: **human annotators and GPT-4o prompted in two different ways**. To estimate the annotation accuracy of each source, we random select 100 samples and verify the labels by human experts. We train each PRM via SFT for 2 epochs, and evaluate results on the AndroidWorld benchmark. Each PRM is evaluated on two criteria: its classification accuracy on a held-out test set, and its downstream impact on task success rate when used as an inference-time verifier. The results are presented in Table 3, showing a direct correlation: as the quality of the annotation data improves, the trained PRM's accuracy improves, which in turn leads to greater gains in task success

| Model | Training Strategy | Training Data | AC High TM↑ | AC High EM↑ |
|-------|-------------------|---------------|-------------|-------------|
| UI-TARS-1.5-7B (Baseline) | - | - | 77.4 | 62.3 |
| +PRM verifier | SFT | Score only | 80.9 | 64.9 |
|  |  | Score + CoT | **81.1** | **65.2** |

Table 4: Comparison of PRMs trained with and without supplementary chain-of-thought data, demonstrating that the inclusion of reason process improves model performance.

| Verifier | Rollouts ($n$) | AC High TM (%)↑ | AC High EM (%)↑ |
|----------|----------------|-----------------|-----------------|
| UI-TARS-1.5-7B (Baseline) | - | 77.4 | 62.3 |
| Actor itself | 3 | 80.8 | 64.8 |
| GUI-Shepherd | 3 | 81.1 | 65.2 |
|  | 5 | 81.7 | 65.8 |
|  | 8 | **81.7** | **65.9** |
|  | 16 | 81.5 | 65.9 |

Table 5: Performance impact of different verifiers and the number of candidate actions ($n$). We compare GUI-Shepherd, against a self-consistency baseline where the actor verifies its actions.

rate. This finding empirically demonstrates the critical importance of a reliable supervisory signal and substantiates our decision to use human annotators for the definitive binary scores.

**Impact of Involving Chain-of-Thought Reasoning.** We conduct a further ablation study to isolate the impact of incorporating chain-of-thought rationales during training. We train two PRM variants for 2 epochs. The first variant, our baseline, is trained via SFT using only the binary correctness scores. The second variant is trained to first generate the corresponding CoT, and then predict the final binary score. We then evaluate two models on their downstream performance as inference-time verifiers. As shown in Table 4, the PRM trained with CoT rationales achieves a notably higher Type Match and Exact Match gain when used for verification. This result suggests that the process of explicitly generating a reasoning chain, acts as a valuable auxiliary task.

**Analysis of the Verification Mechanism.** To further analyze the properties of our verification mechanism, we conduct two targeted experiments. First, we establish a baseline where the actor model serves as its own verifier, using its internal logits to rank its own candidate actions. Our results show that while this self-verification provides a minor improvement, the performance gain from using GUI-Shepherd, is substantially more significant. Second, we investigate the relationship between the number of candidate actions ($n$) and performance. As shown in Table 5, we observe a clear monotonic improvement as we increase the number of candidates from which the PRM selects, despite the higher computational overhead. However, these gains plateau, as performance shows no further improvement when $n$ increases from 8 to 16, suggesting that the benefits have peaked.

## 6 CONCLUSION

In this work, we introduce GUI-Shepherd, a Process Reward Model for reliable, step-by-step GUI agent supervision, trained on a high-quality 52k-example dataset curated via our meticulous pipeline. We demonstrate that GUI-Shepherd significantly enhances agent performance on dynamic, long-sequence GUI tasks by serving as both a dense reward provider for online RL and a reliable inference-time verifier, and show that its benefits extend to offline, single-step action prediction. To our knowledge, this is the first work to systematically explore the effectiveness of PRM in the GUI domain across online RL, offline RL, and inference-time verification. We believe our findings offer a promising path toward developing more capable and generalizable GUI agents.

## 7 ETHICS STATEMENT

This research does not raise any ethical concerns.

## 8 REPRODUCIBILITY STATEMENT

The appendix provides detailed training parameters for both our online and offline RL setups in Appendix C.2 and Appendix D, the methodology for our ARPO reproduction in Appendix C.3, the training details of PRM in Appendix B.3.

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

## A USE OF LLM

**The use of a Large Language Model (LLM) is limited to language editing and refinement.**

## B PRM TRAINING DETAILS

### B.1 STATISTICAL ANALYSIS OF PRM TRAINING DATA

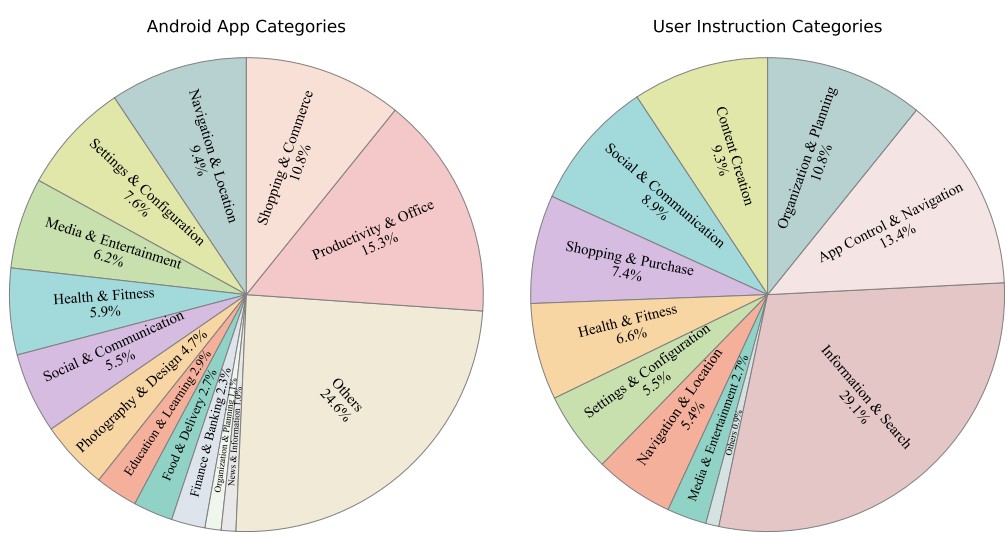

Figure 4: Statistic of PRM training dataset

The Figure 4 above illustrates the distribution of application and user instruction types in our PRM training dataset, whose diversity across various apps and GUI tasks ensures the generalizability of our PRM.

### B.2 DATA ANNOTATION DETAILS

The correctness labels for our PRM are annotated by our in-house data team. As shown in the Figure 5, the annotation interface requires human annotators to determine whether an action in a given state has a positive or negative contribution toward completing the overall instruction. An unsure button is also included, and we filter out any data marked as such during our data curation process. The specific prompt used for GPT-based scoring is detailed in Appendix G.

### B.3 PRM TRAINING PARAMETERS

We initialize our PRM with the UI-TARS-1.5-7B model and fine-tune it for 2 epochs on the training set using Supervised Fine-Tuning. The training is conducted with a constant learning rate of $1 \times 10^{-5}$ and a global batch size of 32 on single H20 node. We formulate the prediction as a binary classification task, where the model generates either a True or False token for the correctness score. Since these two words are encoded as individual tokens in the tokenizer of Qwen2.5-VL (the base model of UI-TARS-1.5-7B), no modifications to the model architecture or training framework are necessary.

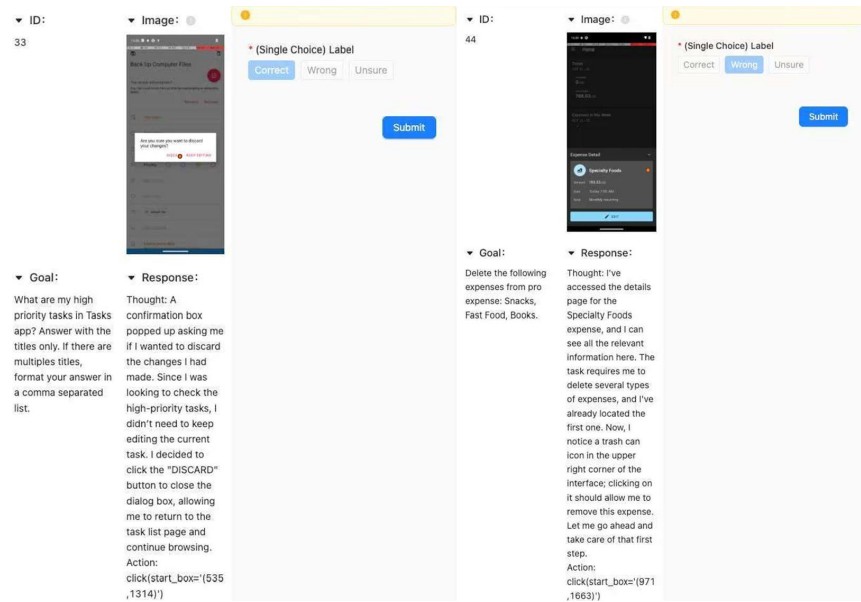

Figure 5: Interface for Human annotation.

## C  ONLINE REINFORCEMENT LEARNING DETAILS

### C.1  REFACTORING THE ANDROIDWORLD BENCHMARK

The original AndroidWorld benchmark relies on the Android_env library as its backend for interacting with the Android emulator. A key limitation of this library is its requirement for direct access to the physical machine, which is incompatible with our company's containerized cluster environment. To address this, we re-engineered the entire interaction layer using the uiautomator2 library. We replaced all of AndroidWorld's Android_env-dependent interfaces for ADB communication, rewrote the functions for retrieving emulator state, and re-implemented all the programmatic rules for evaluating task success or failure.

To ensure the correctness of our new implementation, we first conducted unit tests for every task. To further validate its fidelity, we benchmarked the UI-Tars-1.5-7B model. Since the official results for this model on AndroidWorld have not been released, we compared our results to a reproduction by the GitHub community. Our setup achieved a 32.8% success rate, which is consistent with the community's reported metric, thereby verifying the correctness of our re-implementation.

For our online training setup, we launch an Android emulator instance on a remote machine and expose a dedicated port. This allows us to establish interaction with the emulator via its IP address and the designated port.

### C.2  ONLINE PPO IMPLEMENTATION DETAILS

We train online in parallel with 8 emulators simultaneously. To mitigate the instability of remote connections, which are prone to disconnection, we maintain a pool of 16 emulators, with 8 serving as redundant backups. If the connection to an active emulator is lost, the system automatically switches to a redundant instance to ensure uninterrupted training.

For our PRM-based PPO, we calculate a reward for each step of a completed trajectory. Our PRM is deployed as a vLLM service on a separate node, and we provide these step-wise rewards by calling the service and parsing its output. For the ORM baseline, in contrast, we assign a single terminal reward to the final step of the trajectory based on the programmatic check for task success or failure. Additionally, a "format reward" is computed at each step to penalize unparsable actions. Our online multi-turn PPO is implemented based on the verl codebase.

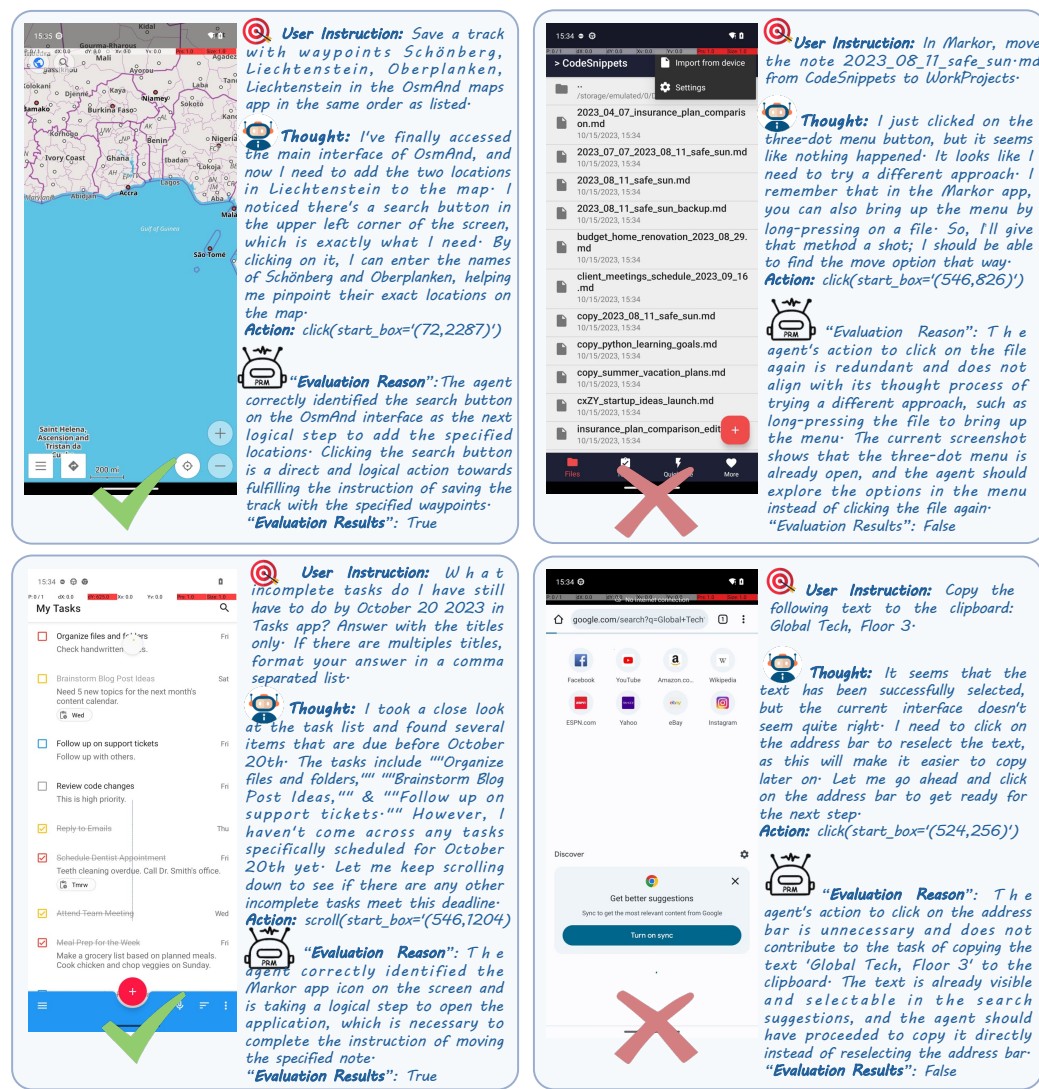

Figure 6: Sample of PRM training data.

The RL training is conducted on a single H20 node. We use a constant learning rate of $1 \times 10^{-6}$ for the actor and $1 \times 10^{-5}$ for the value model, without applying a KL coefficient. In each iteration, we select 8 distinct tasks and collect 8 full trajectories via parallel rollouts on the 8 remote emulators. The maximum number of steps per trajectory is capped at 15. To accommodate the large token size of images, we set the maximum context length to 32k.

## C.3 ARPO REPRODUCTION

We reproduced the ARPO algorithm on the AndroidWorld benchmark. ARPO is an online, trajectory-level GRPO algorithm originally developed for the desktop environment, OSWorld. Our only modification was to replace the logic for interacting with the emulator; the core algorithmic design of ARPO was left unchanged. To align with our PPO experiments, our implementation of ARPO selects one task per iteration and simultaneously rolls out a group of 8 trajectories across 8 emulators.

## C.4 IMPLEMENTATION DETAILS OF VANILLA ORM-BASED PPO

Our ORM-based PPO implementation follows the standard PPO algorithm with **sparse rewards**. The specific mechanism is as follows:

- **Reward Assignment:** For a trajectory $\tau = (s_0, a_0, \ldots, s_T, a_T)$:
  - For all intermediate steps $t < T$, the reward is zero: $r_t = 0$.
  - For the final step $T$, the reward $r_T$ is determined by the rule-based success check: $r_T = 1$ (Success) or $r_T = 0$ (Failure).
- **Training:** The Value Model $V_\phi$ is trained **simultaneously** with the Policy $\pi_\theta$. It learns to estimate the expected return $\mathbb{E}[\sum r]$, thereby providing a per-step value estimation $V(s_t)$ even when the immediate reward is zero.
- **Advantage Calculation:** The sparse rewards $\{r_t\}$ and the learned value estimates $\{V(s_t)\}$ are combined using Generalized Advantage Estimation (GAE) to compute the advantages used for the policy update.

The pseudocode for this ORM-based PPO is provided in Algorithm 3.

---

**Algorithm 3** Vanilla ORM-based PPO (Sparse Reward)

---

Trajectory $\tau = (s_0, a_0, \ldots, s_T, a_T)$ Policy $\pi_\theta$, Value Model $V_\phi$
Initialize reward list $\mathcal{R} \leftarrow []$
**for** $t \leftarrow 0$ $T$ **do** $t < T$ $r_t \leftarrow 0$ *[r]Intermediate steps receive zero reward  *[r]Final step receives Outcome Reward based on rule check CheckSuccess$(\tau) r_t \leftarrow 1$ $r_t \leftarrow 0$ Append $r_t$ to $\mathcal{R}$
Calculate GAE using the learned Value Model Advantages $\leftarrow$ Calculate_GAE$(\mathcal{R}, \text{Values} = V_\phi(\text{States}))$
Update Policy and Value Model Update_Policy$(\pi_\theta, \text{Advantages})$ Update_ValueModel$(V_\phi, \mathcal{R})$

---

Fundamentally, the only implementation difference between our PRM-based PPO and this ORM-based PPO lies in the definition of the reward signal $r_t$:

- **ORM-based PPO:** Uses sparse rewards ($r_t = 0$ until terminal step).
- **PRM-based PPO:** Uses dense process rewards ($r_t = \text{PRM}(s_t, a_t)$ at every step).

All other components (GAE computation, PPO-Clip loss, Value loss) are identical.

## C.5 ADDITIONAL ANALYSIS OF TRAINING SIZE OF PRM

We conducted an additional ablation study on dataset size. We initialized the PRM using UI-TARS-1.5-7B and fine-tuned it using varying subsets of our training data (derived from a 20k dataset). We then evaluated the downstream performance gain when using these resulting models as inference-time verifiers. As illustrated in Table 6, the performance gain during inference yields a steady, monotonic increase as the training data volume grows. Furthermore, the performance of the PRM plateaus once the dataset size reaches a certain scale.

Synthesizing these results with our findings in Table 5, we conclude that the initialization strategy is critical: a strong foundation model inherently possesses a solid start-up capability to serve as a zero-shot reward model. However, the reliability of this verification signal is enhanced as the volume of domain-specific training data increases.

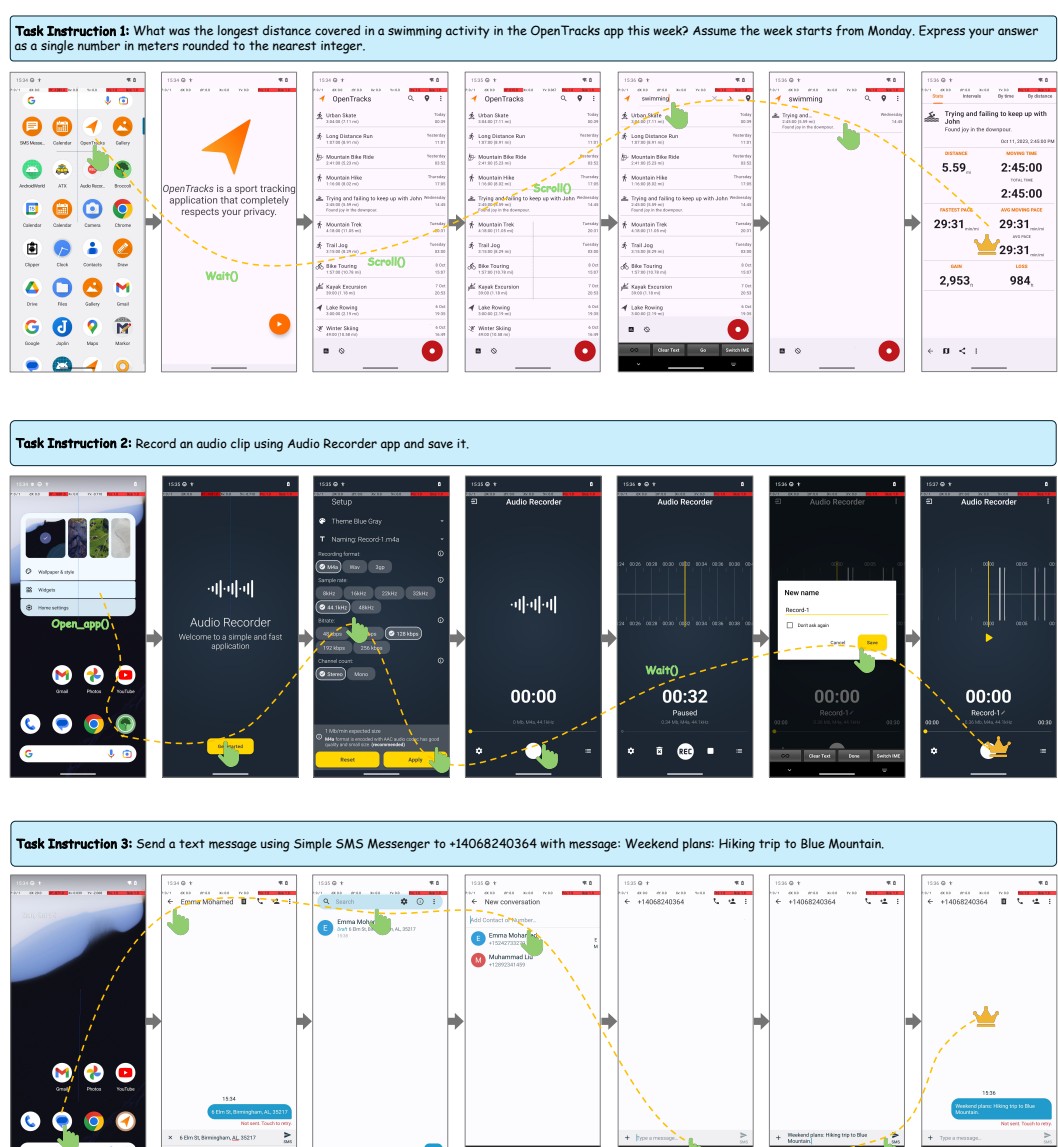

Figure 7: Successful trajectory of our agents.

| Verifier | Params | Data Size | AC-High (TM) | AC-High (EM) |
|----------|--------|-----------|--------------|--------------|
| w/o verifier | 7B | - | 77.4(±0.22) | 62.3(±0.19) |
| UI-TARS-1.5 | 7B | 0 | 80.8(±0.20) | 64.8(±0.19) |
| Trained PRM | 7B | 1k | 80.9(±0.20) | 64.8(±0.20) |
| Trained PRM | 7B | 3k | 81.0(±0.19) | 64.9(±0.19) |
| Trained PRM | 7B | 5k (0.25ep) | 81.0(±0.18) | 65.0(±0.20) |
| Trained PRM | 7B | 10k (0.5ep) | 81.2(±0.19) | 65.3(±0.18) |
| Trained PRM | 7B | 20k (1ep) | 81.4(±0.18) | 65.5(±0.18) |
| Trained PRM | 7B | 40k (2ep) | 81.6(±0.17) | 65.8(±0.18) |
| Trained PRM | 7B | 80k (4ep) | 81.6(±0.17) | 65.7(±0.18) |

Table 6: **Analysis of Data Scaling for PRM Start-up.** We report the performance (Mean ± SEM) on the AC-High benchmark across different training data sizes.

# D PRM PERFORMANCE ON OFFLINE GUI DATASET

To effectively evaluate the efficacy of Process Reward Models in GUI task reinforcement learning, we establish Ground Truth rewards as the oracle upper-bound representing the theoretical performance ceiling achievable with perfect reward signals.

**Reward Design** In our framework, the reward signal is designed to reflect both the correctness of the predicted action and the alignment with textual inputs. For the Oracle reward, we define it as:

$$R_{\text{oracle}} = R_{\text{type}}(y) + \mathbb{I}_{\text{text}}(y) \cdot R_{text}(y)$$

where $y$ is the predicted content of the agent and $R_{\text{type}}(y)$ represents the binary reward based on the action type, indicating whether the predicted action falls within the correct action space. Specifically, for each action $y$, the reward is either 0 or 1 depending on whether it satisfies the required action type. Additionally, if the action type involves text input, we further introduce a matching reward based on the matching score of the input text, $\text{Acc}(y)$, which is also binary (1 for a correct match, 0 for a mismatch).

For the PRM evaluation reward, we use:

$$R_{\text{PRM}} = \mathbb{I}(\text{AC}(y))$$

where $\mathbb{I}(\text{AC}(y))$ is a binary indicator function that provides a reward of 1 if the action $y$ is accepted, and 0 if it is rejected, reflecting the success or failure of the model in producing acceptable actions.

**Implementation Details** We maintain a consistent setup across trials to ensure comparability. The overall batch size is set to 32 with a group size of 8. The KL divergence loss coefficient $\beta$ is fixed at 0.01. During training, we use a constant learning rate of $1 \times 10^{-6}$. The optimizer employed is AdamW with a weight decay of 0.01. All training procedures are conducted on a single node with 8 GPUs.

**Evaluation on AndroidControl** Since the official metrics and evaluation scripts for UI-TARS-1.5-7B on the AndroidControl benchmark have not been released, all reported results for this model were reproduced by our team. We closely followed the official recommendations from the UI-TARS repository's GitHub issues and applied consistent evaluation protocols for all experiments based on UI-TARS-1.5-7B.

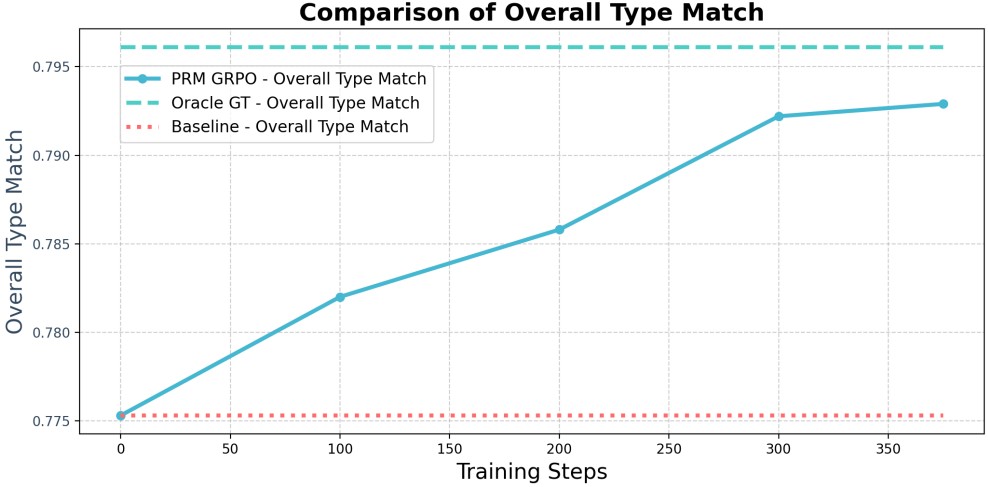

Figure 8: PRM GRPO configuration finally achieves **84.6%** performance gain of oracle reward GRPO upper bound under the same training setting.

| Method | Type Acc | Step Acc | GR Acc | Type Match | Exact Match |
|---|---|---|---|---|---|
| Base | 77.6 | 65.5 | 69.6 | 77.5 | 63.2 |
| UpperBound | 79.7 | 66.5 | 70.1 | 79.6 | 64.1 |
| PRM | 79.4 | 66.3 | 69.9 | 79.3 | 63.9 |

Table 7: Performance on AndroidControl *val* split. UpperBound group adopts $R_{\text{oracle}}$ reward and trained for 200 steps and PRM group adopts $R_{\text{PRM}}$ reported at 375 training steps.

**Results**    Though GT GRPO gains best performance at 200 training steps, examination of extended training trajectories reveals that Process Reward Model demonstrates meaningful capability to approach oracle performance levels despite operating with imperfect reward signals. The PRM-reward configuration shows continuous improvement through 375 training steps, ultimately achieving Overall Type Match of 79.3% and Overall Exact Match of 63.8%. While these metrics remain below the oracle baseline, as is shown in 8, the achievement represents 84.6% of oracle Overall Type Match performance gain and 68.8% of oracle Overall Exact Match performance gain, demonstrating that carefully designed process rewards can closely approximate ground truth supervision quality through sufficient training on static datasets.

# E    MORE ANALYSIS ON PRM

## E.1    USING OPEN SCOURCE MODELS FOR ANNNOTATION

We attempted to utilize the open-source Qwen2.5VL-7B model to annotate training data for the PRM, as illustrated in Figure 9 and Figure 10. We visualized the UI interface, the task description, the annotations from Qwen2.5VL-7B, the human-assigned scores, and the reasoning provided by GPT-4o. Our analysis reveals that Qwen2.5VL-7B suffers from significant hallucination issues; it is easily misled by the erroneous reasoning processes of GUI agents and tends to overlook critical UI elements, leading to incorrect judgments of agent actions. Furthermore, we observed that Qwen2.5VL-7B demonstrates limited capability in instruction following and UI grounding, which further compromises the annotation process. Overall, the annotation quality of the open-source Qwen2.5VL-7B model still lags considerably behind that of human annotators and state-of-the-art closed-source models like GPT-4o.

## E.2    ANALYSIS OF REASONING PROCESS OF PRM

We supplemented our evaluation with a qualitative analysis of the PRM's reasoning process. As shown in Figure 11 and Figure 12, we randomly selected 12 cases and prompted the PRM to analyze the correctness of the GUI Agent's actions based on the given task and state, and then assign a score. The PRM outputs followed the format: $<$Think$>$ Reasoning Process $</$Think$>$ Score: True/False. Observations from these cases indicate that the PRM's reasoning is logically sound; it demonstrates a comprehensive understanding of the task, thoroughly examines the UI screenshots, and rigorously verifies the GUI Agent's thoughts and actions before making a final judgment. This qualitative analysis of the reasoning process further corroborates the reliability of the PRM.

# F    INFERENCE DETAILS

All inference experiments were conducted on NVIDIA H20 nodes utilizing the vLLM backend with bfloat16 precision. The computational environment included flash-attention v2.6.2, vLLM v0.8.5, PyTorch v2.6.0, and CUDA 12.2. For online inference, we employed greedy decoding by setting do_sample to False. Conversely, for offline inference, we enabled sampling with a temperature of 0.8 and top_p of 0.95. We reported the mean and Standard Error of the Mean (SEM) across 5 independent inference runs.

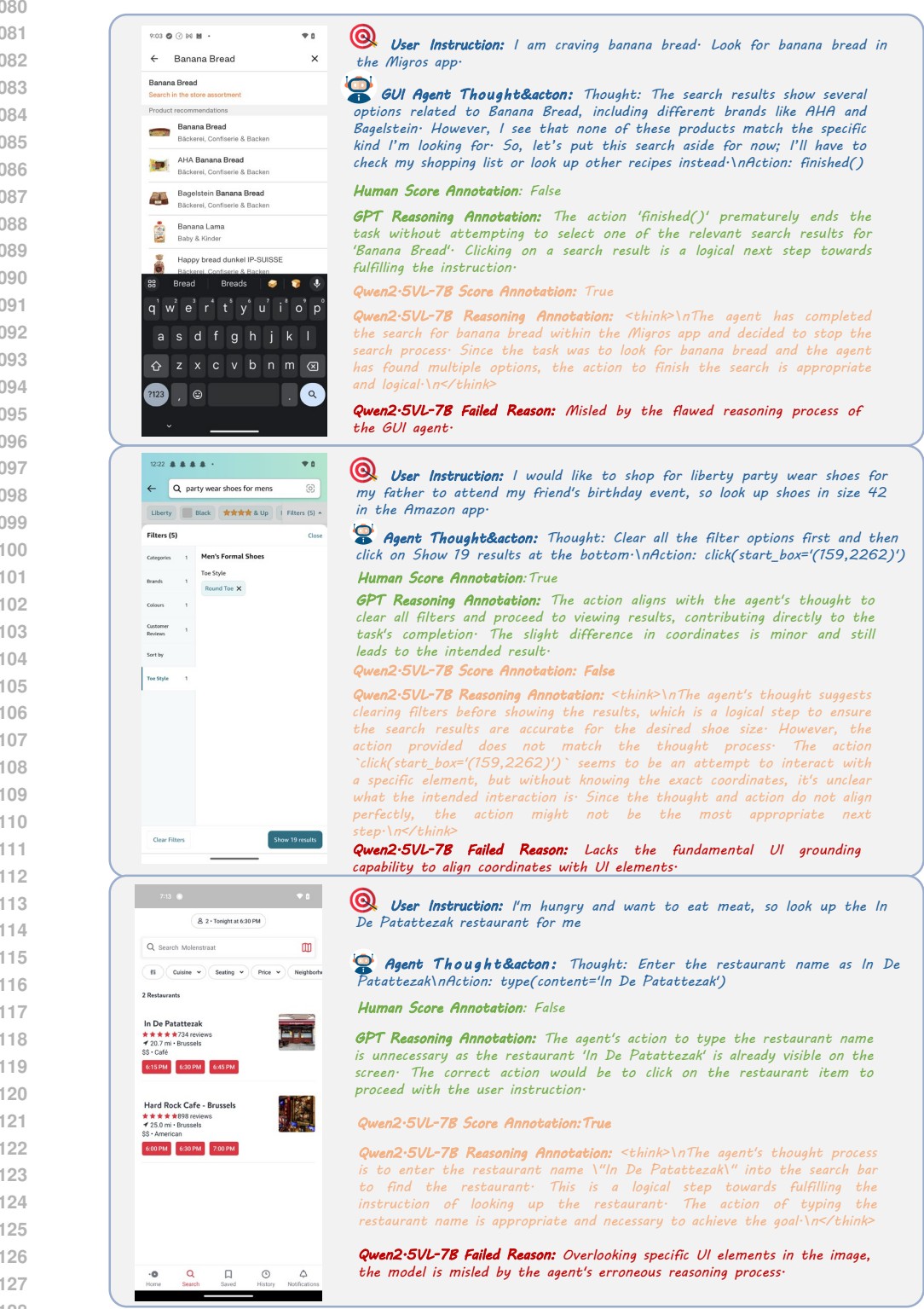

Figure 9: The annotation results from Qwen2.5VL-7B.

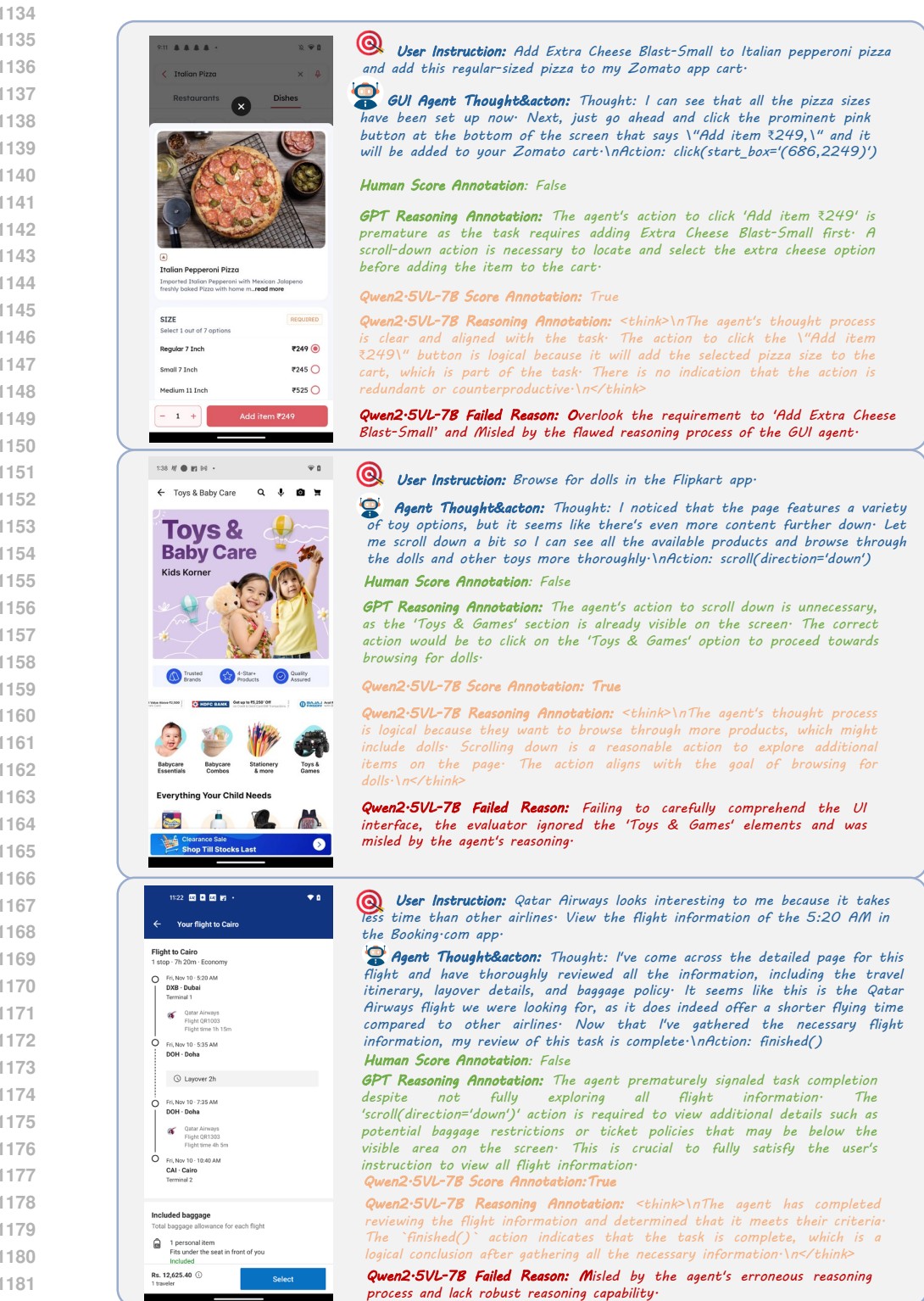

Figure 10: The annotation results from Qwen2.5VL-7B.

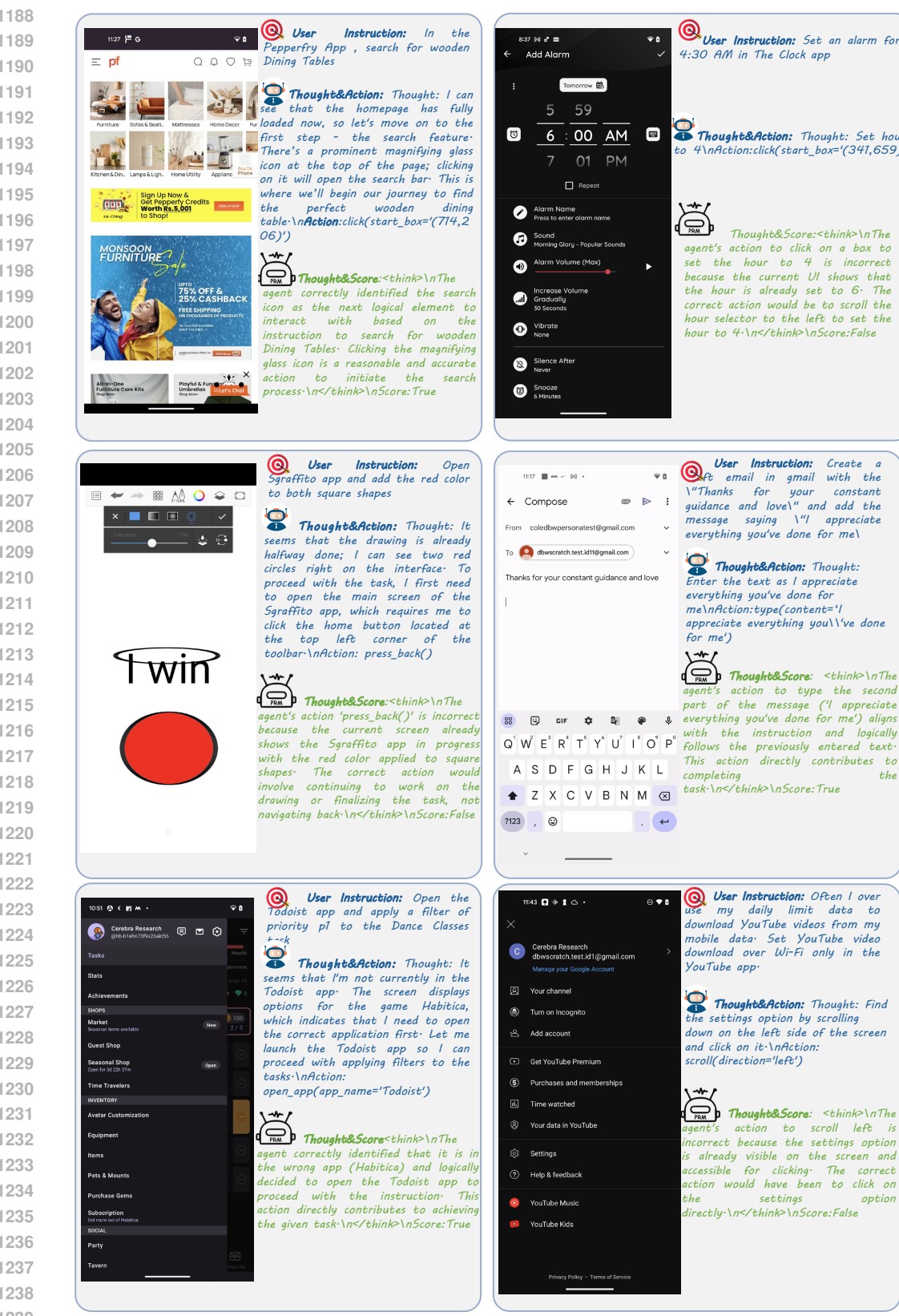

Figure 11: The inference results of PRM.

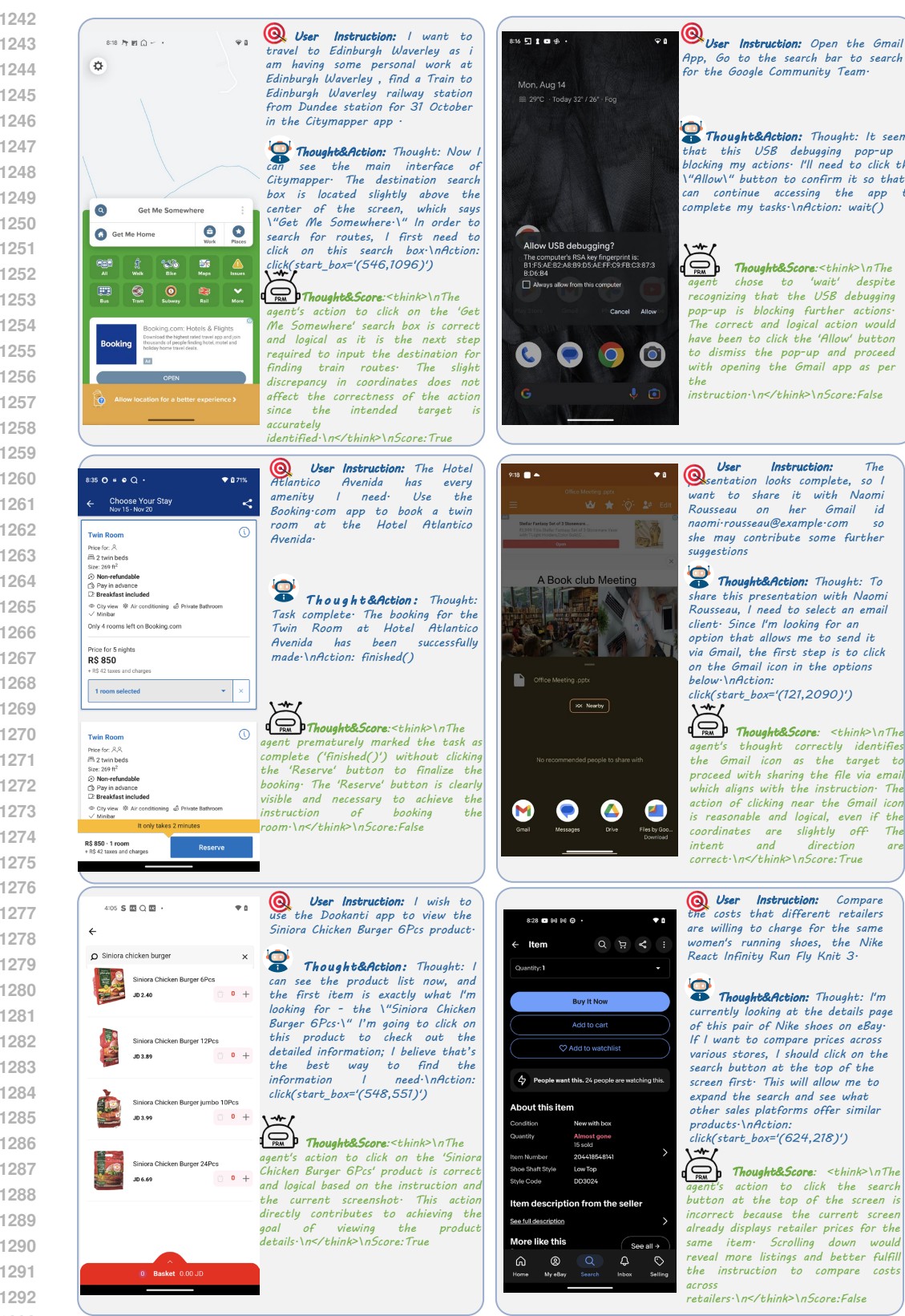

Figure 12: The inference results of PRM.

## G  PROMPT FOR GPT-4O TO GENERATE COT



# Expert Evaluator for an Android GUI Agent

Your mission is to analyze the agent's behavior at each step and determine if its intended action is correct and logical for accomplishing a given task.

### 1. CONTEXT

The agent interacts with a standard Android Operating System. It receives a high-level instruction from a user and attempts to complete it by performing a sequence of actions on the device's GUI.
The process is sequential:

1. The agent observes the current state of the device via a screenshot.
2. Based on the instruction and the screen, it formulates a thought process and decides on a specific action.
3. This action is executed on the device, leading to a new screen state.
4. A screenshot of this new state is captured, and the cycle repeats.

The environment is a live Android OS, which can present real-world challenges like unexpected pop-up dialogs, permission requests, or app onboarding guides.

#### 1.1 AGENT'S ACTION SPACE

The agent's interaction is restricted to the following set of predefined actions. Any action formulated by the agent **must** be one of these types:

- click(start_box='(x1, y1)')
- long_press(start_box='(x1,y1)')
- type(content='text to type')
- scroll(direction='down,up,left,right')
- open_app(app_name='App Name')
- press_home()
- press_back()
- wait()
- finished(content='summary')

### 2. YOUR TASK

You will be provided with four pieces of information for a single step:

1. **Instruction**: The overall goal.
2. **Screenshot**: A PNG image of the current screen.
3. **Agent's Thought and Action**: The agent's reasoning and intended action.
4. **Ground Truth Action**: The expected correct action.

Your task is to critically evaluate the agent's step using the following process:

**A. Understand the Instruction:** Grasp the agent's final objective.

**B. Analyze the History and Screenshot:** Examine the current screen and past actions to assess if the agent is on the right track.

**C. Evaluate the Agent's Action:** Compare the agent's action with your assessment. Consider the Ground Truth Action as background information, but be tolerant with coordinates.

> **IMPORTANT:** NEVER mention phrases like "ground truth" in your reasons. Act as if you don't know the Ground Truth Action.



## 3. EVALUATION CRITERIA

You must use the following criteria to make your judgment:

### Output 1 (Correct / Reasonable Action)

- **Direct Progress:** The action is a clear step towards completing the instruction.
- **Logical Sub-task:** The action is part of a necessary sub-task.
- **Handling Obstacles:** The action correctly solves a UI obstacle.
- **Sensible Exploration:** The action is a reasonable attempt to find necessary controls.
- **Error Correction:** The action appropriately corrects a previous mistake.

### Output 0 (Incorrect / Unreasonable Action)

- **Contradicts Instruction:** The action goes against the goal.
- **Illogical/Impossible Action:** The action makes no sense.
- **Flawed Reasoning:** The agent's thought and action do not match.
- **No Progress / Inefficient:** The action leads nowhere, gets stuck in a loop, or ignores a much more direct path.
- **Perpetuating Errors:** The action continues or compounds a previous error.

## 4. OUTPUT FORMAT

Your response **MUST** be a valid JSON object with **no other text or explanations**. The object must contain two keys: result (integer 1 or 0) and reason (a concise string).

## 5. EXAMPLES

### Example 1: Correctly Handling an Obstacle (Correct)

**Instruction:** Open a new private tab in the browser.
**Agent's Thought and Action:** Thought: A system dialog has appeared asking to set the default browser. This is not required for my current task. I will dismiss this dialog by clicking 'Cancel' to continue. Action: click(start_box='(700, 1300)')
**Expected Output:**

```
"result": 1,
"reason": "The agent correctly identified a system dialog as an obstacle not relevant to the main
     task. Its decision to dismiss the dialog is a logical and efficient way to handle the
     interruption and proceed with the instruction."
```

### Example 2: Correcting a Previous Error (Correct)

**Instruction:** Open the "Clock" app.

**Agent's Thought and Action** Thought: I have accidentally opened the Calculator app... I need to go back to the home screen... Action: press_home()
**Expected Output:**

```
"result": 1,
"reason": "The agent correctly identified its previous error (opening the wrong app) and is taking
     a logical step to correct it by returning to the home screen. This action demonstrates
     appropriate error correction."
```

**Example 3: Error Based on Historical Context (Incorrect)**

**Instruction:** Delete the contact "John Smith".

**Agent's Thought and Action** Thought: A confirmation pop-up has appeared to delete the contact. I need to confirm the deletion to complete the task. Action: click(start_box='(600, 1200)')

**Expected Output:**

"result": 0,
"reason": "The action perpetuates a previous error. The agent is about to delete 'John Appleseed', but the instruction was to delete 'John Smith'. This stems from an incorrect selection in a previous step, and proceeding would fail the task."

**Example 4: Thought and Action Inconsistency (Incorrect)**

**Instruction:** Turn on Wi-Fi.

**Agent's Thought and Action:** Thought: I need to turn on Wi-Fi. I will click the toggle switch next to the 'Wi-Fi' label to enable it. Action: click(start_box='(850, 450)')

**Expected Output:**

"result": 0,
"reason": "The agent's thought process is correct, but its action directly contradicts it. The thought correctly identifies the need to click the Wi-Fi toggle, but the executed action targets the Bluetooth toggle."

**Example 5: Correct and Logical Progress (Correct)**

**Instruction:** Set an alarm for 11:00 AM.

**Agent's Thought and Action:** Thought: I have set the hour to 11. Now I need to set the minutes to 00 as per the instruction. I will click on the '00' on the clock face to select it. Action: click(start_box='(500, 250)')

**Expected Output:**

"result": 1,
"reason": "The action is a correct and logical step towards fulfilling the instruction. The agent has correctly identified the next required input (setting the minutes to '00') and is clicking the correct UI element to do so."

## 6. HISTORICAL CONTEXT

The full history of preceding actions is provided below. Review this sequence to understand the agent's journey so far, which is crucial for identifying if the agent is correcting a mistake or continuing a flawed path.

## 7. YOUR TURN

This is the historical actions of the Agent to complete the task, and the screenshot after each action.

