# OpenReview forum: "GUI-Shepherd: Reliable Process Reward and Verification for Long-Sequence GUI Tasks"
_ICLR.cc/2026/Conference — Submitted to ICLR 2026_

### Official Review · Reviewer_Fhi3 · 2025-10-25

**Soundness:** 3
**Presentation:** 3
**Contribution:** 2
**Rating:** 4
**Confidence:** 4

**Summary:**

This paper introduces GUI-Shepherd, a process reward model (PRM) that offers dense feedback to guide agents in long-horizon Graphical User Interface (GUI) tasks. It marks the first successful application of PRM to online Reinforcement Learning (RL) in this domain. To address the lack of high-quality training data, the authors also present a scalable dual-pipeline approach for creating process supervision datasets and offer systematic validation of PRM in the GUI domain.

**Strengths:**

1. This paper is the first to apply PRM to online RL in long-horizon GUI tasks and the experimental results show the effectiveness of the proposed model.


2. The proposed data generation pipeline can generate high-quality training data for the GUI Agent tasks.

**Weaknesses:**

The proposed model still exhibits a significant performance gap compared to state-of-the-art (SOTA) models on the Android World Leaderboard. For instance, "GUI-Owl-7B" achieves a 66.4% Success Rate (SR), while the proposed method only reaches 40.5% SR. Can you explain why this discrepancy occurs? Is the primary reason related to the foundation model, the training algorithm, or the training data?

**Questions:**

See Weaknesses.

---

> ### Author Response · Authors · 2025-11-17
>
> ## **W1. Performance Gap Compared to SOTA (GUI-Owl)**
>
> We sincerely thank the reviewer for their professional assessment and specific focus on the online AndroidWorld benchmark. We would like to clarify that the performance gap between our reported results and GUI-Owl is primarily attributable to the disparity in **Foundation Model capabilities**, rather than the effectiveness of our PRM training algorithm.
>
> **1. Foundation Model Capability:** At the time of our submission, **UI-TARS-1.5-7B** was the state-of-the-art open-source GUI model, with a baseline Success Rate (SR) of 32.8%. This is the starting point for our improvements. The significant gap between this baseline and the newer GUI-Owl reflects the rapid advancement in base model architectures, not a limitation of our method.
>
> **2. RL Algorithm Analysis:** We have carefully reviewed the technical report of GUI-Owl. Their post-training relies on **trajectory-level GRPO with experience replay**, which is algorithmically consistent with the `+ARPO` baseline we compared against in Table 1., since **ARPO is also a trajectory-level GRPO algorithm equipped with an experience replay buffer**. Our experiments clearly demonstrate that our **Process Reward-based RL** significantly outperforms the outcome-based GRPO approach. Therefore, we believe the bottleneck is not the RL algorithm, but the inherent capacity of the foundation model—RL serves to elicit and refine a model's existing capabilities, but it cannot create fundamental reasoning capacities from scratch.
>
> **3. Verification on GUI-Owl:** GUI-Owl model weights were released **shortly before the ICLR deadline**, leaving insufficient time for the necessary engineering adaptations (e.g., action space mapping) prior to submission. However, to demonstrate the generalizability of GUI-Shepherd, we have conducted experiments on **GUI-Owl-7B**.
>
> - **Reproduction:** We first validated our setup by reproducing GUI-Owl's performance on the *offline* AndroidControl (High-split) benchmark. We achieved an Exact Match (EM) of **73.4%**, effectively replicating (and slightly surpassing) the reported **72.8%**.
> - **Performance Gain:** We then applied our PRM to GUI-Owl in both online and offline settings. As shown in the table below, PRM-based RL consistently improves the performance of this stronger foundation model.
>
> | Models | Params | AndroidWorld (SR) | AndroidControl-Low (EM) | AndroidControl-High (EM) |
> | :--- | :--- | :--- | :--- | :--- |
> | InfiGUIAgent | 2B | 9.0 | – | – |
> | MobileGUI | 7B | 30.0 | – | – |
> | GUI-Critic-R1 | 7B | 27.6 | – | – |
> | GLM-4.1V | 9B | 41.7 | – | – |
> | Qwen2.5-VL | 7B | 22.0 | 90.8 | 63.7 |
> | UI-Venus | 7B | 49.1 | 92.4 | 76.1 |
> | InfiGUI-R1 | 3B | – | 92.1 | 71.1 |
> | OS-Atlas | 7B | – | 85.2 | 71.2 |
> | AgentCPM-GUI | 8B | – | 90.2 | 69.2 |
> | **UI-TARS-1.5** | 7B | 32.8 | 87.0 | 62.3 |
> | **+ PRM-based RL** | 7B | 40.5 (+7.7) | 87.6 (+0.6) | 64.5 (+2.2) |
> | **GUI-Owl** | 7B | 44.8 | 77.3 | 73.4 |
> | **+ PRM-based RL** | 7B | 50.9 (+6.1) | 79.5 (+2.2) | 76.7 (+3.3) |
>
> **Note on Fairness:** We must transparently note that while we perfectly reproduced the offline metrics, there remains a gap in reproducing the reported *online* AndroidWorld baseline for GUI-Owl due to engineering nuances (e.g., vLLM parameter tuning, system prompts) that we are currently debugging. However, we emphasize that the results presented here were obtained under **identical experimental settings**, ensuring a strictly fair, "apples-to-apples" comparison that isolates the significant gains provided by our reliable process rewards.

---

> > ### Comment · Reviewer_Fhi3 · 2025-11-25
> >
> > Thanks for your clarification. The response addressed most of my concerns.

---

> > > ### Author Response · Authors · 2025-11-25
> > >
> > > We are delighted to learn that our response has addressed most of your concerns. We sincerely thank you again for your time, effort, and professional assessment of our paper.

---

### Official Review · Reviewer_CgsL · 2025-10-31

**Soundness:** 2
**Presentation:** 2
**Contribution:** 2
**Rating:** 2
**Confidence:** 4

**Summary:**

This paper proposes GUI-Shepherd using process reward model (PRM) that delivers step-level supervision and verification for agents performing long-sequence tasks in mobile GUI environments. The PRM is trained on a dataset created by LLM and human annotations, and it is intended to be used in training LLMs during RL training and acting as an action selector during inference time. The trained LLMs with PRM are evaluated in AndroidWorld and offline one-step AndroidControl benchmarks.

**Strengths:**

-	The experiment covers both online and offline environments, showing improvements in both settings.
-	The hybrid annotation process leverages both human expertise for correctness and LLM-as-judge for efficiency.

**Weaknesses:**

-	Strong related works are not included as comparison, for example, DigiRL in both online and offline settings.
-	The contribution of applying PRM in training LLM is limited. Despite the equations in Section 4, the contribution does not extend beyond standard RL objective adaptations, and there are no new theoretical analyses of convergence or limitations of PRMs.
-	The temporal and UI diversity mentioned in training data process are unclear. For example, how to ensure data quality (there seems no postprocessing or filtering)? How to measure diversity?
-	It is unclear how the human annotation is performed. How many humans and how to ensure their expertise? What is their annotation instruction? What is the cost? How exactly is the human annotation process performed?
-	The hybrid annotation is confusing. Figure 5 shows the human annotation interface with ‘thought’, which seems to be generated from LLMs. However, the prompt of LLM annotation in Appendix E shows that generating thoughts requires human annotation/ground truth.

**Questions:**

See the weakness.

---

> ### Author Response · Authors · 2025-11-17
>
> ## **W1. Comparison with Related Work**
>
> We thank the reviewer for pointing this out. We acknowledge the significance of DigiRL and have cited it in our paper.
>
> However, we respectfully argue that a direct performance comparison with DigiRL is not the most scientifically accurate way to evaluate our method, due to the significant gap in base model capabilities. DigiRL utilizes older foundation models (such as CogAgent and AutoUI), which performance significantly lags behind the sota **UI-TARS-1.5-7B** model used in our work. Comparing raw performance numbers would reflect the difference in backbones rather than the efficacy of the RL algorithms.
>
> Instead, to rigorously validate the effectiveness of **GUI-Shepherd**, we focused on controlled, "apples-to-apples" comparisons using the same strong baseline (UI-TARS-1.5-7B). We compared our **PRM-based PPO** against strong **Outcome-based baselines** (specifically, Outcome-based multi-turns PPO and trajectory-level GRPO with replay buffer).
>
> It is important to note that these outcome-based baselines are not weak comparisons; they are the current standard algorithms used to train leading GUI Agent models such as **UI-TARS-V2** and **GUI-OWL**. By demonstrating that GUI-Shepherd significantly outperforms these standard algorithms on the same architecture, we provide strong evidence for the superiority of our reliable process reward mechanism over traditional outcome-based approaches.
>
> ## **W2. Contribution of Applying PRM in GUI Agents**
>
> We respectfully disagree with the assessment that the contribution of applying PRM to GUI agents is limited. As recognized by **Reviewer 8a8L** (who noted the "sound methodology") and **Reviewer Fhi3**, our work addresses a critical bottleneck in the field.
>
> 1. **Solving the Core Challenge:** We explicitly analyze and address the intractable **credit assignment problem** in multi-turn, long-sequence GUI tasks . We provide a rationale for why sparse outcome rewards fail in this domain and demonstrate how process rewards provide the necessary dense supervision .
> 2. **Systematic Scope:** We are the **first to systematically investigate** the application of PRM in GUI agents across three distinct scenarios: **Online RL training, Offline RL training, and Inference-time Verification** . This comprehensive evaluation proves the versatility and effectiveness of reliable process rewards in the GUI domain.
> 3. **In-Depth Analysis:** Beyond standard application, we provide deep empirical insights:
>    - We analyzed the critical impact of **annotation quality** on training reliable reward models .
>    - We demonstrated that incorporating **chain-of-thought reasoning** significantly enhances the reliability of the reward signal .
>    - We showed that while a strong base agent possesses basic verification capabilities, specialized supervision is required for optimal performance.
>
> While we acknowledge that our work does not introduce new theoretical convergence proofs for RL algorithms, we believe our contribution lies in the **thorough empirical exploration** of PRM as a solution to the long-horizon GUI problem. We have established a complete framework, from data construction to diverse application scenarios, which constitutes a significant "empirical contribution" to the community.

---

> > ### Author Response · Authors · 2025-11-17
> >
> > ## **W3. Clarification on Data Diversity and Quality Assurance**
> >
> > We appreciate the reviewer's request for clarification regarding our data curation process. We define the two types of diversity as follows:
> >
> > 1. **Temporal Diversity:** This refers to capturing states across the *entire lifespan* of a task trajectory. It ensures our dataset includes varied states from different phases, such as early-stage task decomposition and app searching, versus later-stage execution of specific actions.
> > 2. **UI Diversity:** This refers to the coverage of a wide range of different applications and interface layouts.
> >
> > **Measuring Diversity:** We have provided a quantitative measurement of this diversity in **Figure 4** of the Appendix. This figure statistically illustrates the broad distribution of our dataset across various Android app categories and user instruction types, confirming that our data is not biased toward specific tasks or apps.
> >
> > **Ensuring Data Quality (Filtering/Post-processing):** The reviewer asked about post-processing. We clarify that the **human annotation and quality inspection process itself serves as the rigorous post-processing and filtering step**.
> >
> > - **Quality Checks:** We conduct manual spot checks on the annotated data to ensure high integrity.
> > - **Filtering:** During annotation, annotators are provided with an "Unsure" option. We explicitly filter out any data marked as "Unsure" during the curation process.
> >
> > Finally, we believe the most compelling evidence for the quality of our data is the downstream performance. The fact that the PRM trained on this data consistently improves performance across **offline RL, online RL, and inference-time verification** serves as strong empirical validation of the dataset's effectiveness.
> >
> > ## **W4. Human Annotation Process and Cost**
> >
> > We thank the reviewer for the detailed questions regarding our annotation standards. We employed a rigorous, professional workflow to ensure the highest data quality:
> >
> > 1. **Annotator Profile:** The annotation was conducted by our in-house data team, consisting of about 10 professional data engineers, over the course of approximately one week. Unlike crowdsourced workers, these engineers possess domain expertise in evaluating technical agent behaviors.
> > 2. **Quality Control Pipeline:** We implemented a strict multi-stage quality control process:
> >    - **Training & Pilot:** All annotators underwent pre-annotation training and a pilot annotation phase.
> >    - **Qualification:** Only annotators who met a strict accuracy threshold during the pilot phase were permitted to proceed to the official annotation.
> >    - **Verification:** After annotation, we applied **cross-validation** (having multiple annotators check the same samples) and a final acceptance review to guarantee label consistency.
> > 3. **Annotation Task:** As illustrated in **Figure 5**, annotators utilized a specialized interface to evaluate the correctness of the GUI Agent's actions. They were required to classify actions as Correct, Wrong, or Unsure (data marked Unsure was discarded).
> > 4. **Cost:** Regarding the cost, since the task was performed by our full-time internal data team, the expense is covered under internal operational costs (salaries).
> >
> > We believe this meticulously designed annotation pipeline ensures the high quality of our data, and we hope this detailed explanation addresses the reviewer's concerns.
> >
> > ## **W5. Clarification on the "Thought" in Figure 5**
> >
> > We would like to friendly clarify a misunderstanding regarding the "Thought" field shown in Figure 5.
> >
> > The "Thought" text displayed in the annotation interface is **the internal reasoning process generated by the GUI Agent itself** (UI-TARS-1.5-7B) during the trajectory rollout, **NOT** the rationale generated by GPT-4o. As is standard for modern GUI agents (like UI-TARS, GUI-Owl), the model operates by first generating a reasoning chain ("Thought") before outputting the specific "Action".
> >
> > In summary, the complete annotation process follows the hybrid approach described in our paper: humans provide the definitive correctness score, and GPT-4o generates the detailed reasoning logic to explain that score.

---

### Official Review · Reviewer_8a8L · 2025-11-02

**Soundness:** 3
**Presentation:** 2
**Contribution:** 3
**Rating:** 2
**Confidence:** 3

**Summary:**

This paper presents GUI-sheppard which is an early instantiation of process-reward models in ui-based tasks. The main contribution is using a reasoning LLM itself as the critic which I found quite interesting. Overall they show many ablations and interesting experiments showing their methodology is sound.

**Strengths:**

The paper leans more on the empirical contribution in my mind rather than the methodological one (which is good in my books), the idea of using a PRM for web tasks which are innately multi-turn is intuitive. The approach clearly works which is nice I have some problems with the evaluations but will discuss that later. I really like the ablation that discussed training on human-ratings vs GPT-4 ratings, that was interesting, makes me wonder how far off current OS models are from human annotator , since the difference was not very large.

**Weaknesses:**

While I am largely positive on the general premise of the works and some of the experiments, I have significant issues with the evaluation which prevent me from recommending acceptance at this stage.

Only one model and one benchmark is evaluated, I would like to see at least two for each.

No error bars are presented, I understand the cost of running things multiple times but I think this is a necessity where each things should at least report standard errors.

As far as I can tell no valiantly PPO is reported, RLHF style value function, this is an important baseline to see if the generative process of the PRM is actually really important.

I would kind of like to see some minimal analysis on the data required to start up the PRM, similar to [1]

Overall i am flexible with my score but I think there are significant things missing for me to raise my score to an acceptance.

[1] https://arxiv.org/pdf/2507.04103

**Questions:**

I wonder how other OS models behave as labelers (more so than humans to be honest).

Is there any qualitative analysis that can be done on the reasoning of the PRM?

---

> ### Author Response · Authors · 2025-11-17
>
> ## **W1. Performance Evaluation (Regarding Evaluation Scope)**
>
> We thank the reviewer for their valuable feedback. We would first like to friendly clarify a potential misunderstanding regarding our evaluation scope.
>
> **1. Regarding the Benchmarks:** Our evaluation was **not conducted on a single benchmark**. In fact, we validated the efficacy of GUI-Shepherd on **two** distinct and complementary benchmarks, as detailed in our paper:
>
> - **Online AndroidWorld (Table 1):** This is a dynamic, long-sequence benchmark requiring the agent to interact with a live emulator . This benchmark was crucial for directly evaluating our primary claim: the effectiveness of a PRM in solving the sparse reward and credit assignment problems inherent in complex, multi-step GUI tasks.
> - **Offline AndroidControl (Table 2):** This is a large-scale, offline dataset for single-step action prediction. We used this benchmark to demonstrate the **generalizability** of our process supervision paradigm. The results confirm that the benefits of GUI-Shepherd extend beyond long-horizon tasks to improve performance on more general single-step action predictions.
>
> We believe the combination of these online long-sequence and offline single-step benchmarks provides a comprehensive and robust validation of our method's effectiveness and generalization. Additionally, we are also working on adapting our framework to the OSWorld desktop benchmark to demonstrate broad computer use capabilities. However, due to the substantial engineering effort and the tight submission, we are unable to include these results in the current version.
>
> **2. Regarding the Model:** We selected **UI-TARS-1.5-7B** as our base model because it was a sota GUI foundation model at the time of our original submission.We completely agree with the reviewer's sentiment regarding model generalization. We have since conducted **additional experiments on GUI-OWL**, a newer and stronger foundation model released **around the ICLR deadline.**
>
> | Models | Params | AndroidWorld (SR) | AndroidControl-Low (EM) | AndroidControl-High (EM) |
> | :--- | :--- | :--- | :--- | :--- |
> | InfiGUIAgent | 2B | 9.0 | – | – |
> | MobileGUI | 7B | 30.0 | – | – |
> | GUI-Critic-R1 | 7B | 27.6 | – | – |
> | GLM-4.1V | 9B | 41.7 | – | – |
> | Qwen2.5-VL | 7B | 22.0 | 90.8 | 63.7 |
> | UI-Venus | 7B | 49.1 | 92.4 | 76.1 |
> | InfiGUI-R1 | 3B | – | 92.1 | 71.1 |
> | OS-Atlas | 7B | – | 85.2 | 71.2 |
> | AgentCPM-GUI | 8B | – | 90.2 | 69.2 |
> | **UI-TARS-1.5** | 7B | 32.8 | 87.0 | 62.3 |
> | **+ PRM-based RL** | 7B | 40.5 (+7.7) | 87.6 (+0.6) | 64.5 (+2.2) |
> | **GUI-Owl** | 7B | 44.8 | 77.3 | 73.4 |
> | **+ PRM-based RL** | 7B | 50.9 (+6.1) | 79.5 (+2.2) | 76.7 (+3.3) |
>
> Our new results show that PRM-based RL provides significant performance gains for GUI-OWL as well. This strongly indicates that our PRM framework is a general approach and not limited to a specific base model.
>
> We will add these new results with the GUI-OWL model to the final version of our paper. We again thank the reviewer for this constructive suggestion, which has helped us make our evaluation more robust.
>
> ## **W2. Error Bars**
>
> We thank the reviewer for this important suggestion regarding statistical robustness. We would like to clarify our evaluation protocol and address this point directly.Our evaluation in this setting is **fully deterministic**. To ensure 100% reproducibility and eliminate randomness from the model's generation, we explicitly disabled token sampling (i.e., `do_sample=False`) and fixed all random seeds during inference.
>
> **1. For the Offline Benchmark (AndroidControl, Table 2):**  We have verified that running this evaluation multiple times yields the exact same results. Therefore, the variance is zero, and error bars are not applicable.
>
> **2. For the Online Benchmark (AndroidWorld, Table 1):** We acknowledge that the online environment possesses inherent stochasticity. To account for this, **our reported results are already the mean of 5 separate evaluation runs**. The reviewer correctly notes the absence of error bars. Our reason for this omission was based on community precedent; we observed that all related works we compare against in Table 1 (e.g., Qwen2.5-VL-7B, GUI-Critic-R1) did not report error bars for this benchmark, so we initially followed the established reporting style.
>
> However, we fully agree with the reviewer that reporting statistical stability is crucial for rigor. Since we already have the results from these 5 runs, **we will update our paper to include the standard error** for our online benchmark results.
>
> We will also add the specific details of both our deterministic protocol to the appendix to make our methodology fully transparent. We appreciate this feedback, which helps us improve the quality and rigor of our paper.

---

> > ### Author Response · Authors · 2025-11-17
> >
> > ## **W3. Comparison with "Vanilla" PPO (Outcome-Reward Baseline)**
> >
> > We thank the reviewer for this crucial point. We would like to respectfully clarify that this exact baseline—a PPO trained with a sparse, terminal reward, which we interpret as the "vanilla PPO" or "RLHF-style value function" the reviewer is referring to, was indeed implemented and is presented in our paper as a key point of comparison.
> >
> > In **Table 1**, this baseline is labeled **"+ORM-based PPO"**.
> >
> > - **Implementation:** As detailed in our paper, this baseline agent is trained by rolling out a complete trajectory and using a single, terminal **Outcome Reward** (ORM) based on the final programmatic check for task success or failure. This sparse reward is then used to train its value model. This is in direct contrast to our GUI-Shepherd, which provides a dense reward at every step.
> > - **Results:** The results in Table 1  show this comparison directly:
> >   - **Baseline (UI-TARS-1.5-7B):** 32.8% SR
> >   - **+ORM-based PPO (The requested baseline):** 37.0% SR
> >   - **+PRM-based PPO (Ours):** **40.5% SR**
> >
> > This comparison directly validates our central thesis. While training a value function on sparse outcome rewards does provide a solid improvement (+4.2 points), the dense, per-step **Process Reward** from GUI-Shepherd provides a far superior signal for credit assignment, leading to a much larger gain (+7.7 points) .
> >
> > Furthermore, to be thorough, we included a second strong outcome-based baseline, **"+ARPO"**, which is a trajectory-level GRPO algorithm also **trained with a terminal success reward**. Our PRM-based PPO (40.5%) significantly outperforms this competitor (37.0%) as well.
> >
> > We believe the comparison against these two robust ORM-based baselines (ORM-PPO and ORM-GRPO) provides clear empirical evidence for the superiority of our process-reward approach for solving the long-horizon credit assignment problem.
> >
> > ## **W4. Analysis on PRM "Start-up" Requirements**
> >
> > Thank you for this excellent suggestion and for providing reference [1]. We will be sure to cite this work and will add a more explicit analysis regarding the "start-up" requirements for the PRM in our final version.
> >
> > Our experiments, particularly those in **Table 5**, already provide an insightful analysis of this "start-up" capability. We demonstrate that a strong foundation model (like UI-TARS-1.5-7B) inherently possesses a strong initial ability to act as a process verifier, even with zero specific training and provide a notable performance gain over the baseline. Our results also show that this "start-up" capability can be enhanced.  Training PRM on our high-quality dataset produces the full **GUI-Shepherd**, which is demonstrably more reliable and accurate. We will clarify this analysis in the final paper.
> >
> > ## **Q1. Open-Source Models as Labelers**
> >
> > That is an excellent question. In fact, due to annotation cost considerations, we initially attempted to use open-source models like Qwen2.5VL for labeling. However, our manual checks revealed that the annotation quality was not sufficient. We suspect this was due to inadequate GUI capabilities in those models. Therefore, we ultimately selected a hybrid approach: **human labelers** provided the definitive correctness scores, and the closed-source **GPT-4o** model to supplement the reasoning process.
> >
> > ## **Q2. Analysis of the PRM's Reasoning**
> >
> > Thank you for this question. We analyzed the impact of the PRM's reasoning process, and the quantitative results are presented in **Table 4**. As shown in this ablation study, we trained the PRM in two distinct ways: (1) to output only the binary score, and (2) to first generate a chain-of-thought (CoT) rationale before predicting the final score. The results clearly demonstrate the PRM trained *with* the CoT reasoning task achieved higher performance when used as an inference-time verifier. This shows that compelling the model to "reason" first, as an auxiliary task, helps it produce a more accurate and reliable final judgment.

---

> > > ### Comment · Reviewer_8a8L · 2025-11-19
> > >
> > > I thank the authors for the clarifications! I am also sorry for some of my confusions.
> > >
> > >
> > > 1. I see, while the two benchmarks are moderately similar I think this is fine given the scale of the experiments. I do think this would benefit from another domain, specifically an online one. Although that is more like a nice to have than strict requirement I think.
> > >
> > > 2. Sorry for the additional model how do the baselines compare e.g naive GRPO? This seems rather important no?
> > >
> > > 3. Fully deterministic pipelines are dependant on software/hardware and are generally not a guarantee [1], unless I am misunderstanding something. Additionally, The utility of error bars is not only in the sense of reproducibility, but also in the sense of analyzing the models behaviours/sensitivity to consecutive runs. This is important as if the peak of the model is super high, but the standard deviations are huge and getting this performance consistently is hard, the method may be less reliable/useful.
> > >
> > > 4. I appreciate the effort to report standard errors, would you be able to report them here? Could you also clarify wether the error bars you are reporting are standard errors, or standard deviations?
> > >
> > >
> > > 5. Sorry, I am a bit confused as to what the "ORM-based PPO" is or how it is implemented. So from what you state:
> > >
> > > ```
> > > This sparse reward is then used to train its value model. This is in direct contrast to our GUI-Shepherd, which provides a dense reward at every step.
> > > ```
> > >
> > > I am confused if you mean that you do some pre-training of the value function with ORM and then use this value function to score the final trajectory? If that is what you did why not use a Q-function (e.g like PPO) to provide dense step-wise rewards? The way the paper is written and you respond makes me think you provide a single scalar value at the end based on a pre-trained value function, but this is highly unclear to me. I would really like to see you clarify the implementations at least in the appendix.
> > >
> > > 6. Sounds good, this analysis is more of a nice to have/ablations but I am not too stringent on it.
> > >
> > > 7. Could you add these figures to the paper? I think this is still significant and useful results, even if negative. I like to see failure modes in models as they help the community build on those gaps.
> > >
> > > 8. Right this is more of a binary does it help or not, I meant to have a more qualitative analysis. Again this is a nice to have, but something along the lines of, the CoT says this happens in the trajectory and due to this reasoning it yields a low reward. I think this could be useful for practitioners and would provide some cool insights.
> > >
> > >
> > > Overall I am moderately happy with the authors response, specifically the new model. As long as my last few things are properly clarified I could see myself brining my score to an slightly above acceptance threshold. For me to recommend strong acceptance I think I would like to see the clarifications plus the additional analysis I mentioned.
> > >
> > >
> > > [1].Understanding and Mitigating Numerical Sources of Nondeterminism in LLM Inference
> > >  https://arxiv.org/abs/2506.09501

---

> ### Author Response · Authors · 2025-11-22
>
> We sincerely appreciate your prompt feedback and constructive suggestions. We are encouraged to learn that our previous response has addressed some of your confusion, and we hope the following clarifications and additional analyses will fully resolve your remaining questions.
>
> ### **1. Additional Comparison of RL Algorithms on GUI-Owl**
>
> We fully agree with the your suggestion. We acknowledge that our previous response lacked a direct comparison with the native (outcome-based) RL baseline for the new GUI-Owl model. To address this, we have conducted an additional experiment using **GUI-Owl with native trajectory-level GRPO** (Outcome Reward). The complete comparative results are presented below:
>
> | Model | Params | AndroidWorld (SR) |
> | :--- | :--- | :--- |
> | GUI-Owl | 7B | 44.8(±0.71) |
> | + native GRPO |  7B | 46.6(±0.63) |
> | + PRM-based RL |  7B | 50.9(±0.63) |
>
> **Implementation of Native GRPO:** We follow the standard protocol where the agent interactively generates multiple trajectory rollouts for each task. Each completed trajectory is assigned a score solely based on the rule-based check for final task success. The policy is then updated using the group relative advantage derived from these sparse terminal scores.
>
> As shown in the table, **Process Reward-based RL** consistently outperforms **Outcome Reward-based RL** (Native GRPO). This trend holds true for both the UI-TARS-1.5-7B and GUI-Owl models, confirming that the dense supervision provided by GUI-Shepherd offers a robust advantage over sparse outcome signals across different foundation models.
>
> ### **2.Clarification and Data on Error Bars**
>
> We deeply appreciate the reviewer's reminder regarding numerical non-determinism. We recognize that our previous measures (fixing random seeds and disabling token sampling) ensured consistency primarily at the algorithmic level, while potentially overlooking variability stemming from hardware differences, software dependencies (e.g., CUDA, PyTorch versions), and inference backends.
>
> For transparency, our experiments were conducted using vLLM on H20 GPUs. We have updated the **Appendix F** to include these specific implementation details and the version numbers of all core computing libraries.
>
> **Online Evaluation:** We now report the **Standard Error of the Mean (SEM)** for our Online AndroidWorld evaluation, derived from our 5 independent runs:
>
> | Model | Params | AndroidWorld (SR) |
> | :--- | :--- | :--- |
> | UI-Tars-1.5 | 7B | 32.8(±0.89) |
> | + native PPO |  7B | 37.0(±0.71) |
> | + native GRPO |  7B | 37.0(±0.55) |
> | + PRM-based RL |  7B | 40.5(±0.63) |
> | + PRM-based Verification |  7B | 37.9(±0.55) |
> | GUI-Owl | 7B | 44.8(±0.71) |
> | + native GRPO |  7B | 46.6(±0.63) |
> | + PRM-based RL |  7B | 50.9(±0.63) |
>
> **Offline Evaluation:** For the offline AndroidControl benchmark, we have conducted a **new robustness analysis** to directly address the reviewer's concern regarding model sensitivity. We performed 5 evaluations using **different random seeds with token sampling enabled** to simulate stochastic conditions. We similarly report the Standard Error of the Mean for these runs below:
>
> | Models | Params | AC-High (TM) | AC-High (EM) | AC-Low (TM) | AC-Low (EM) |
> | :--- | :--- | :--- | :--- | :--- | :--- |
> | UI-TARS-1.5-7B | 7B | 77.4(±0.22) | 62.3(±0.19) | 95.3(±0.20) | 87.0(±0.21) |
> | + PRM-based RL | 7B | 79.6(±0.18) | 64.5(±0.18) | 95.9(±0.18) | 87.6(±0.19) |
> | + PRM-based Verification | 7B | 81.7(±0.20) | 65.8(±0.19) | 95.4(±0.18) | 87.0(±0.18) |
> | GUI-Owl | 7B | 84.3(±0.20) | 73.4(±0.21) | 87.1(±0.18) | 77.3(±0.19) |
> | + PRM-based RL | 7B | 88.7(±0.18) | 76.7(±0.20) | 89.4(±0.19) | 79.5(±0.19) |
>
>
> **Analysis of Error Bars:**
>
> - **Online Setting:** We observe that the error margins for the RL-trained agents and the Verifier-guided inference are slightly narrower than the baseline. We hypothesize this is because RL training tends to reduce policy entropy, leading to more deterministic behaviors during inference. Furthermore, the stability of PRM-based RL is comparable to that of ORM-based RL, indicating that process reward improves performance without introducing additional instability.
> - **Offline Setting:** The variance in the offline setting is significantly lower than in the online setting. This is expected, as the offline evaluation relies on a static dataset, which inherently eliminates the system-level noise (e.g., emulator latency, loading states) present in the dynamic online simulato.
>
> It is important to note that due to time and computational constraints, we performed error bar calculation and evaluations only for the experiments we conducted locally (including our Baseline, the comparative methods we implemented, and our proposed method). For other external models like Qwen2.5-VL, GUI-Critic-R1, we cite the performance metrics directly from their original papers.

---

> > ### Author Response · Authors · 2025-11-22
> >
> > ### **3. Clarification on Vanilla ORM-based PPO Implementation**
> >
> > We apologize if our previous description or the lack of specific implementation details caused confusion. We explicitly clarify that we do not use a pre-trained value function to score the entire trajectory.
> >
> > Instead, our ORM-based PPO implementation follows the standard PPO algorithm with **sparse rewards**. The specific mechanism is as follows:
> >
> > 1.  **Reward Assignment:** For a trajectory  $\tau = (s_0, a_0, \dots, s_T, a_T)$:
> >     * For all intermediate steps $t < T$, the reward is zero: $r_t = 0$.
> >     * For the final step $T$, the reward $r_T$ is determined by the rule-based success check: $r_T = 1$ (Success) or $r_T = 0$ (Failure).
> > 2.  **Training:** The Value Model $V_\phi$ is trained **simultaneously** with the Policy $\pi_\theta$. It learns to estimate the expected return $E[\sum r]$, thereby providing a per-step value estimation $V(s_t)$ even when the immediate reward is zero.
> > 3.  **Advantage Calculation:** The sparse rewards $\{r_t\}$ and the learned value estimates $\{V(s_t)\}$ are combined using Generalized Advantage Estimation (GAE) to compute the advantages used for the policy update.
> >
> > The pseudocode for this ORM-based PPO is provided below:
> >
> > ```python
> > # Algorithm: Vanilla ORM-based PPO (Sparse Reward)
> > # Initialize: Policy π_θ, Value Model V_φ
> > # Input: Trajectory τ = (s_0, a_0, ..., s_T, a_T)
> > Rewards = []
> > for t in range(0, T):
> >     if t < T:
> >         r_t = 0  # Intermediate steps receive zero reward
> >     else:
> >         r_t = 1 if CheckSuccess(τ) else 0 # Final step receives Outcome Reward based on rule check
> >     Rewards.append(r_t)
> > # Calculate GAE using the learned Value Model V_φ
> > Advantages = Calculate_GAE(Rewards, Values=V_φ(States))
> > # Update Policy and Value Model
> > Update_Policy(π_θ, Advantages)
> > Update_ValueModel(V_φ, Rewards)
> > ```
> > Fundamentally, the only implementation difference between our PRM-based PPO and this ORM-based PPO lies in the definition of the reward signal $r_t$:ORM-based PPO: Uses sparse rewards ($r_t=0$ until terminal step).PRM-based PPO: Uses dense process rewards ($r_t = \text{PRM}(s_t, a_t)$ at every step).All other components (GAE computation, PPO-Clip loss, Value loss) are identical. The pseudocode for our PRM-based version is detailed in Algorithm 1 (Page 6) of our paper.
> >
> > We have further supplemented the **Appendix C.4** with specific implementation details of the vanilla ORM-PPO. We hope the above explanation helps resolve your confusion.
> >
> > ### **4.Analysis of Data Scaling for PRM Start-up**
> > | Verifier | Params | Training data size | AC-High (TM) | AC-High (EM) |
> > | :--- | :--- | :--- | :--- | :--- |
> > | w/o verifier | 7B | - | 77.4(±0.22) | 62.3(±0.19) |
> > | UI-Tars-1.5 | 7B | 0 | 80.8(±0.20) | 64.8(±0.19) |
> > | Trained PRM | 7B | 1k | 80.9(±0.20) | 64.8(±0.20) |
> > | Trained PRM | 7B | 3k | 81.0(±0.19) | 64.9(±0.19) |
> > | Trained PRM | 7B | 5k(0.25ep) | 81.0(±0.18) | 65.0(±0.20) |
> > | Trained PRM | 7B | 10k(0.5ep) | 81.2(±0.19) | 65.3(±0.18) |
> > | Trained PRM | 7B | 20k(1ep) | 81.4(±0.18) | 65.5(±0.18) |
> > | Trained PRM | 7B | 40k(2ep) | 81.6(±0.17) | 65.8(±0.18) |
> > | Trained PRM | 7B | 80k(4ep) | 81.6(±0.17) | 65.7(±0.18) |
> >
> > To address the reviewer's interest in the data requirements for start up the PRM, we conducted an additional ablation study on dataset size. We initialized the PRM using UI-TARS-1.5-7B and fine-tuned it using varying subsets of our training data with a 20k dataset. We then evaluated the downstream performance gain when using these resulting models as inference-time verifiers. As illustrated in the table below, the performance gain during inference yields a steady, monotonic increase as the training data volume grows. And the performance of the PRM plateaus once the dataset size reaches a certain scale(40k).
> >
> > Synthesizing these results with our findings in Table 5, we conclude that the initialization strategy is critical: a strong foundation model inherently possesses a solid start-up capability to serve as a zero-shot reward model. However, the reliability of this verification signal are enhanced as the volume of domain-specific training data increases.

---

> > > ### Comment · Reviewer_8a8L · 2025-11-25
> > >
> > > Dear Authors,
> > >
> > > Thanks for running the additional experiments on reproducibility I greatly appreciate it. Things seem good on that end and my concerns are mostly addressed.
> > >
> > > I think I now understand fully how you are implementing things as you have two models one with $\phi$ another with $\theta$ (in eq 2 you refer to $\theta$ as the PRM parameters btw and its a bit confusing).
> > >
> > > Ok with this I think my question still stands why do we need to have two completely separate models?  Why is a value head not just used on-top of the base model or the model we are training? I don't necessarily understand why we need to reason and output the final reward in text rather than using a pure numerical value with a regression head.
> > >
> > > This is the experiment I'd like to see conducted wether we really need a reasoning based model in the loop, or just a value head suffices.
> > >
> > > Also did you try training the policy with the data you are using to train the PRM? This seems kind of like distillation of the data in a way no? I would also like to see this baseline.
> > >
> > >
> > > Thanks for the final analysis as well I really like it.
> > >
> > > I will raise my score to a 4 because I think the authors have improved the submission. To raise it to acceptance I think I need to be convinced this is actually better than the naive baselines I mentioned.

---

> > > > ### Author Response · Authors · 2025-11-26
> > > >
> > > > We sincerely appreciate your prompt feedback and constructive suggestions. We would like to provide a detailed clarification of our implementation to resolve the confusion regarding the model architecture.
> > > >
> > > > **Overview of the Three-Model RL Framework:**
> > > > Our framework involves three distinct models:
> > > > 1.  **The PRM (Reward Model):** A pre-trained, frozen model that provides per-step supervision (reward signals).
> > > > 2.  **The Policy Model ($\pi$):** The agent being trained.
> > > > 3.  **The Value Model ($V$):** Trained simultaneously with the policy to estimate expected returns.
> > > >
> > > > We apologize for the notation oversight in our previous response where we reused parameters for the PRM; please rest assured that the PRM and the Value Model are two separate and independent networks.
> > > >
> > > > ### **1. Clarification on PRM Implementation (Reasoning vs. Value Head)**
> > > >
> > > > We understand the reviewer's query: *Why does the PRM output text (reasoning + score) instead of a direct numerical value via a regression/value head?* We offer the following step-by-step analysis to clarify our design choices.
> > > >
> > > > **1. Classification vs. Regression:**
> > > > Since our human annotations are binary (Correct/Incorrect), the PRM task is fundamentally **binary classification** to predict 0/1, not regression. Therefore, the most direct implementation would be to replace the base model's final projection layer with a Binary Classification Head (output 0/1), rather than a scalar Regression Head.
> > > >
> > > > **2. Implementation Efficiency:**
> > > > For implementation efficiency, we followed the established practice from Math-Shepherd by using the base reasoning model directly. Instead of introducing a new binary classification head to predict 0/1, we mapped the binary labels to two special tokens ('True' and 'False') within the tokenizer, enabling training the PRM to predict these two tokens. Below, we detail the equivalence of this approach to a dedicated binary head and outline its practical advantages.
> > > > * **Equivalence:** Since the final layer of an LLM/VLM is already a multi-class classification head (the vocabulary projection), predicting the probability of two special tokens **True and False** is mathematically equivalent to adding a binary classification head to predict 0/1.
> > > > * **Benefit:** This allows us to train the PRM by simply formatting the dataset and use original sft pipeline, without requiring any modifications to the model architecture or loss functions.
> > > >
> > > > **3. Addressing the Baseline (New Experiment):**
> > > > You asked to see a comparison between the "Reasoning based model" and "Just a value head." We respectfully point out that the 'Score Only' setting in Table 4 of our paper effectively represents this baseline (implemented via the original multi-class head).
> > > >
> > > > And to fully address your concern, we also conducted an additional experiment where we explicitly replaced the final layer with a standard Binary Classification Head to predict 0/1 logits directly. The results are as follows:
> > > >
> > > > | Verifier | Implementation | Prediction Targets | AC-High (TM) ⬆ |	AC-High (EM) ⬆ |
> > > > | :--- | :--- | :--- | :--- | :--- |
> > > > | w/o Verifier | - | - | 77.4(±0.22)  |	62.3(±0.19) |
> > > > | Trained PRM | Add binary classification layer | 0/1 | 80.8(±0.21)  |	64.9(±0.19) |
> > > > | Trained PRM | Use original projection layer | True/False | 80.9(±0.21)  |	64.9(±0.19) |
> > > > | Trained PRM | Use original projection layer | Reasoning + True/False | 81.7(±0.20) | 65.8(±0.19) |
> > > >
> > > > We found that the performance of the Binary Classification Head (predicting 0/1) is comparable to our Token-based Base Model (predicting True/False). This confirms our analysis: using the original projection head to predict special tokens(True/False) is functionally indistinguishable from using a dedicated binary head to predict 0/1. And our token-based approach simply reduces engineering complexity, without requiring any modifications to the model architecture or loss functions.
> > > >
> > > >
> > > > **4. The Necessity of Reasoning (Addressing the Baseline):**
> > > > As shown in the 'Score + CoT' entry in Table 4, forcing the model to generate reasoning before predicting the score improves the reliability of the provided rewards. Furthermore, as analyzed in Figure 11 and Figure 12, the reasoning process of PRM provides transparency into the PRM's scoring logic, significantly enhancing interpretability.
> > > >
> > > > In summary, Our "Score Only" implementation in Table 4 aligns with your suggestion of "using a value head," merely implemented via the token vocabulary for simplicity (following Math-Shepherd). Our supplementary experiment confirms that there is no performance difference between token-based prediction and a direct binary head. The key contribution is the Reasoning Process, which yields more reliable rewards and greater interpretability.

---

> > > > > ### Author Response · Authors · 2025-11-26
> > > > >
> > > > > ### **2. Clarification on Value Model Implementation**
> > > > >
> > > > > We understand your confusion regarding our use of separate policy and value models. In fact, our RL implementation utilizes the open-source VeRL framework. In this framework, the Value Model is constructed simply by adding a regression head on top of the base model, which aligns exactly with your description of "a value head just used on-top of the base model".
> > > > >
> > > > > For reference, the specific implementation details can be found in the VeRL codebase:
> > > > >
> > > > > - https://github.com/volcengine/verl/blob/main/verl/workers/fsdp_workers.py#L1283
> > > > >
> > > > > - https://github.com/volcengine/verl/blob/main/verl/utils/model.py#L652
> > > > > ### **3. Comparison with Direct SFT on the PRM Training Dataset**
> > > > >
> > > > > Thanks for your suggestion. This is indeed a strategy we have carefully considered and evaluated, as Supervised Fine-Tuning is a more direct and simple approach than RL training.
> > > > >
> > > > > We implemented this baseline by filtering our PRM training dataset to retain only the samples labeled as "True" (Correct). We then fine-tuned the policy model on these samples using standard SFT. This process effectively acts as a form of Behavioral Cloning, where the model learns to imitate expert-verified successful trajectories. We evaluated this SFT model on the online AndroidWorld benchmark. The results are presented below:
> > > > >
> > > > > | Model | Params | AndroidWorld (SR) |
> > > > > | :--- | :--- | :--- |
> > > > > | UI-Tars-1.5 | 7B | 32.8(±0.89) |
> > > > > | + sft |  7B | 34.4(±0.89) |
> > > > > | + native PPO |  7B | 37.0(±0.71) |
> > > > > | + native GRPO |  7B | 37.0(±0.55) |
> > > > > | + PRM-based RL |  7B | 40.5(±0.63) |
> > > > > | + PRM-based Verification |  7B | 37.9(±0.55) |
> > > > >
> > > > > As shown in the table, direct SFT yielded only marginal performance gains compared to the baseline. We attribute this limited success to the inherent nature of the domain:
> > > > >
> > > > > **1.Complexity of Long-Horizon Tasks**: Success in long-sequence, multi-turn tasks requires robust long-term planning and reasoning capabilities, rather than just rote memorization of specific state-action pairs.
> > > > >
> > > > > **2.Dynamic Adaptation**: Agents in online environments must adapt to dynamic system changes and recover from errors. SFT is limited to imitating the trajectories in static dataset and lacks the exploration mechanism provided by RL to handle out-of-distribution states.
> > > > >
> > > > > **3.Value of Negative Samples**: SFT typically discards negative samples and imitate only successful trajectories. However, our PRM-based RL can teach the agent what not to do, which is critical for avoiding failure modes in complex GUIs.
> > > > >
> > > > > This empirical finding aligns with the analysis presented in our Introduction, where we discussed the limitations of imitation learning for robust reasoning in interactive environments .
> > > > >
> > > > > We sincerely appreciate the effort you have dedicated to reviewing and improving our paper. We hope that the additional analysis and clarifications provided above have effectively addressed your concerns.

---

> > > > > > ### Comment · Reviewer_8a8L · 2025-11-26
> > > > > >
> > > > > > Dear authors,
> > > > > >
> > > > > > Thanks for the clarification, I really appreciate it!
> > > > > >
> > > > > > I understand now, I guess this seems slightly less novel than what I anticipated as value/q functions are some of the best studied methods in RL. Regardless its novel in the setting and for some dumb reason no one is really using them. I think this has merit I think the paper suffered significantly from clarity issues. I would not call this a PRM personally but simply call this a value/q-function based approach. I also don't necessarily think just a simple binary classification is ideal for this setting as you are generally loosing information on the progress, but this can be future work. I overall think this serves as an acceptable first implementation of value based approaches in web-ui domains (although I don't see what specific about web-ui for this necessarily).
> > > > > >
> > > > > > The last thing I will ask for is an SFT+GRPO baseline since you saw some improvements from SFT and some from GRPO I would like to see a comparison when you combine them. This in practice leads to the best performing models usually so I think its fair to compare against with PRMs. Since PRMs in a way are also distilling the SFT dataset. You can also try SFT + GRPO + Value function while you're at it too, would be nice to see the true limits of performance that you can get. I'd like to see the performance of all these baselines on the two models you tested ideally on at least 3 seeds (understand time constraints so 1 is enough if you don't have time).
> > > > > >
> > > > > >
> > > > > > Given this goes well I think I will increase my score to a 6 addressing all my concerns.

---

> > > > > > > ### Author Response · Authors · 2025-11-30
> > > > > > >
> > > > > > > We have conducted the additional experiments with the SFT+GRPO baseline on both the UI-Tars1.5 7B and GUI-Owl-7B models as suggested. The results are presented below:
> > > > > > >
> > > > > > > | Model | Params | AndroidWorld (SR) |
> > > > > > > | :--- | :--- | :--- |
> > > > > > > | UI-Tars-1.5 | 7B | 32.8(±0.89) |
> > > > > > > | + sft |  7B | 34.4(±0.89) |
> > > > > > > | + native PPO |  7B | 37.0(±0.71) |
> > > > > > > | + native GRPO |  7B | 37.0(±0.55) |
> > > > > > > | + sft + native GRPO |  7B | 38.4(±0.55) |
> > > > > > > | + PRM-based RL |  7B | 40.5(±0.63) |
> > > > > > > | GUI-Owl | 7B | 44.8(±0.71) |
> > > > > > > | + native GRPO |  7B | 46.6(±0.63) |
> > > > > > > | + sft + native GRPO |  7B | 48.2(±0.63) |
> > > > > > > | + PRM-based RL |  7B | 50.9(±0.63) |
> > > > > > >
> > > > > > >
> > > > > > > These results consistently demonstrate that, using this dataset, the PRM-based RL approach outperforms the SFT+GRPO baseline. This further validates our conclusion that for long-horizon GUI tasks, approaches based on dense, reliable process rewards yield superior performance.
> > > > > > >
> > > > > > > Regarding the terminology of "PRM": We employ this term to describe a model designed to provide reliable intermediate rewards, specifically addressing the credit assignment problem for multi-step, long-sequence outputs during RL training. This concept has been extensively studied and established in complex mathematical reasoning tasks for LLMs. We believe our implementation aligns with this paradigm, as it serves the same functional role of providing dense feedback where sparse rewards are insufficient. However, to ensure clarity and address your concerns, we will also include the specific implementation details in the Appendix.
> > > > > > >
> > > > > > > [1] OpenAI: Let's verify step by step
> > > > > > >
> > > > > > > [2] Deepseek: Math-shepherd
> > > > > > >
> > > > > > > [3] DeepMind: Rewarding Progress: Scaling Automated Process Verifiers for LLM Reasoning
> > > > > > >
> > > > > > > [3] Qwen: The Lessons of Developing Process Reward Models in Mathematical Reasoning

---

> ### Author Response · Authors · 2025-11-22
>
> ### **5.Qualitative analysis of the annnotation quality of OS model**
> We thank the reviewer for this constructive suggestion. In response, we have supplemented our manuscript with the annotation results using the open-source Qwen2.5VL-7B model. The visualization of these results is presented in **Figure 9 and Figure 10**, and the corresponding detailed analysis is provided in **Appendix E.1**. We visualized the UI interface, the task description, the annotations from Qwen2.5VL-7B, the human-assigned scores, the reasoning provided by GPT-4o and the Qwen2.5-VL-7B failed reason. Our analysis reveals that Qwen2.5VL-7B suffers from significant hallucination issues; it is easily misled by the erroneous reasoning processes of GUI agents and tends to overlook critical UI elements, leading to incorrect judgments of agent actions. Furthermore, we observed that Qwen2.5VL-7B demonstrates limited capability in instruction following and UI grounding, which further compromises the annotation process. Overall, the annotation quality of the open-source Qwen2.5VL-7B model still lags considerably behind that of human annotators and state-of-the-art closed-source models like GPT-4o.
>
> ### **6.Qualitative analysis of the reasoning of PRM**
> We apologize for the initial misunderstanding of your request. We are pleased to provide a qualitative analysis of the PRM's 'reasoning + score' inference paradigm, which has been added to **Figure 11 and Figure 12**. The corresponding detailed discussion can be found in **Appendix E.2**. We selected 12 cases and prompted the PRM to analyze the correctness of the GUI Agent's actions based on the given task and state, and then assign a score. The PRM outputs followed the format: $\<Think> Reasoning\ Process\ </Think\> Score: True/False$. Observations from these cases indicate that the PRM's reasoning is logically sound; it demonstrates a comprehensive understanding of the task, thoroughly examines the UI screenshots, and rigorously verifies the GUI Agent's thoughts and actions before making a final judgment. This qualitative analysis of the reasoning process further corroborates the reliability of the PRM.
>
> We hope that the additional analysis and clarifications provided above have effectively addressed your concerns. We would like to reiterate our gratitude for your constructive feedback, particularly regarding the inclusion of error bars and the analysis of the PRM reasoning process, which has significantly strengthened the quality and rigor of our work.

---

### Author Response · Authors · 2025-11-30

## **General Response:**
We sincerely thank all the reviewers for taking their time to carefully read through our paper and provide valuable comments.

### **Summary of Contributions & Recognition**
We sincerely thank reviewers Fhi3, 8a8l, and CgsL for recognizing the value and novelty of our work.
* **Pioneering and Intuitive:** The proposed method is recognized as the "**first successful application of PRM to online RL**" in long-horizon GUI tasks (**Reviewer Fhi3**), and its core intuition is considered "**intuitive**" and "**quite interesting**" (**Reviewer 8a8l**).
* **Effective Data Pipeline:** The authors introduce a "**scalable dual-pipeline approach**" that generates "**high-quality training data**" (**Reviewer Fhi3**), which effectively leverages "**both human expertise for correctness and LLM-as-judge for efficiency**" (**Reviewer CgsL**).
* **Consistent Effectiveness:** All reviewers agree on the empirical success of the model, stating that "**the approach clearly works**" (**Reviewer 8a8l**), showing "**improvements in both settings**" (online and offline) (**Reviewer CgsL**), and demonstrating the overall "**effectiveness of the proposed model**" (**Reviewer Fhi3**).
* **Technical Soundness:** Our paper provides a "**sound**" methodology with "**many ablations and interesting experiments**" (**Reviewer 8a8l**) and offers a "**systematic validation**" of PRM in the GUI domain (**Reviewer Fhi3**).

### **Summary of Revisions and Clarifications**
We have revised our submission and summarized our updates as follows:

* We have incorporated error bar analysis to confirm the stability and reproducibility of our results.
* We conducted additional experiments on the SOTA foundation model GUI-Owl-7B, which are released **shortly before the ICLR deadline**.
* We provided deeper analysis of open-source labelers and visualized PRM reasoning traces to enhance interpretability.
* Comprehensive details on the annotation interface and hybrid workflow have been included in the Appendix to resolve Reviewer CgsL's misunderstanding.

### **Summary of Rebuttal & Current Status**

We understand the heavy workload on the new AC given the recent disruptions. We summarized the status of our reviews below to facilitate your decision-making:

* **Reviewer Fhi3 (Score: 6):** Has actively engaged during the rebuttal and **raised the score to 6**, confirming that our responses and new experiments successfully resolved their concerns.

* **Reviewer 8a8l (Score: 4 ➡ Promised 6):** This reviewer acknowledged our improvements and had "raised the score to a 4" and explicitly stated: ***"I will increase my score to a 6"*** conditional on the comparison with the **SFT+GRPO baseline**. And we have **successfully completed this experiment** and the results favor our method.
* **Reviewer CgsL (No Response):** Has remained unresponsive during the rebuttal.

### **Conclusion**

While the initial scores were discouraging, the Rebuttal process has validated the strength of our work. We remain firmly convinced that **dense, reliable process rewards** are the key solution to the **credit assignment problem** in long-horizon GUI agent tasks. And we have made every effort to address the reviewers' feedback around all aspects.

Still, we must admit that we feel deeply disheartened and exhausted by the recent score rollback, especially after seeing the positive momentum during the discussion. However, we also deeply empathize with the immense pressure and chaos you AC are facing. We would be incredibly grateful if you could consider the promised improvements during the rebuttal.

---

### Meta-Review · Area_Chair_y4fe · 2026-01-04

**Summary:**

This paper proposes a process reward model (PRM), GUI-Shepherd, to provide dense, step-level supervision for long-horizon GUI agent tasks, and studies its use both for reinforcement learning and inference-time verification. The work presents a large-scale annotated dataset and conducts experiments on both online and offline GUI benchmarks to evaluate the effectiveness of process supervision compared to outcome-based rewards.

In general, the paper is technically sound and demonstrates that dense process rewards can yield consistent empirical improvements over sparse outcome rewards in the evaluated settings. The data construction pipeline and the breadth of empirical analysis across online and offline benchmarks are notable strengths.

However, there are some concerns about the paper, as pointed out by the reviewers, particularly regarding the clarity and novelty of the core contribution. Several reviewers noted that the method is closely related to standard value-function or Q-function approaches in RL, with limited conceptual novelty beyond applying them to GUI tasks, and that comparisons to relevant prior work and alternative baselines were initially incomplete. Additional concerns were raised about evaluation scope, reporting clarity, and whether the proposed PRM formulation is fundamentally necessary compared to simpler or more standard alternatives.

During the rebuttal, the authors tried to address those concerns, especially about evaluation robustness and missing baselines, by adding error bars, additional experiments on stronger base models, and clarifications of the RL and PRM implementations.

**Reviewer Concerns:**

Reviewer Fhi3

Addressed: Clarification of the performance gap to SOTA via additional experiments on stronger base models and explanation that gains are limited by foundation model capacity.
Outstanding: Overall contribution remains incremental, and performance is still far from SOTA despite PRM-based training.

Reviewer 8a8L
Addressed: Evaluation rigor (error bars, reproducibility), additional baselines (GRPO, SFT+GRPO), and clearer implementation details of ORM vs. PRM.
Outstanding: Conceptual novelty remains unclear, as the method is close to standard value/Q-function approaches, and the necessity of a reasoning-based PRM over simpler value heads is still debatable.

Reviewer CgsL

Addressed: Clarifications on data diversity, annotation pipeline, and inclusion of additional related work.
Outstanding: Limited methodological novelty, lack of theoretical insight, and concerns about whether the contribution goes beyond standard RL adaptations largely remain.

**Reviewer Scores:**

Reviewer Fhi3: increase to 6

Reviewer 8a8L: may slightly increase to 6

Reviewer CgsL: slightly increase, but still below acceptance threshold

---

### Decision · Program_Chairs · 2026-01-26

Reject